# Response outcome gates the effect of spontaneous cortical state fluctuations on perceptual decisions

**Davide Reato\*†, Raphael Steinfeld†, André Tacão-Monteiro, Alfonso Renart\***

Champalimaud Research, Champalimaud Foundation, Lisbon, Portugal

**Abstract** Sensory responses of cortical neurons are more discriminable when evoked on a baseline of desynchronized spontaneous activity, but cortical desynchronization has not generally been associated with more accurate perceptual decisions. Here, we show that mice perform more accurate auditory judgments when activity in the auditory cortex is elevated and desynchronized before stimulus onset, but only if the previous trial was an error, and that this relationship is occluded if previous outcome is ignored. We confirmed that the outcome-dependent effect of brain state on performance is neither due to idiosyncratic associations between the slow components of either signal, nor to the existence of specific cortical states evident only after errors. Instead, errors appear to gate the effect of cortical state fluctuations on discrimination accuracy. Neither facial movements nor pupil size during the baseline were associated with accuracy, but they were predictive of measures of responsivity, such as the probability of not responding to the stimulus or of responding prematurely. These results suggest that the functional role of cortical state on behavior is dynamic and constantly regulated by performance monitoring systems.

## Editor's evaluation

Reato and colleagues investigated a question that has long puzzled neuroscientists: what features of ongoing brain activity predict trial-to-trial variability in responding to the same sensory stimuli? The data demonstrate that taking into account behavior on the previous trial (specifically an incorrect choice) allows these associations to be seen. This is an important advance in our understanding of the relationship between brain state, behavioral state, and performance. Technically, the study is convincing, with appropriate and validated methodology in line with current state-of-the-art.

**\*For correspondence:**
davide.reato@neuro.fchampalimaud.org (DR);
alfonso.renart@neuro.fchampalimaud.org (AR)

†These authors contributed equally to this work

**Competing interest:** The authors declare that no competing interests exist.

## Introduction

Successfully performing any behavior, including the acquisition and processing of sensory information to guide subsequent action, requires that the dynamical regimes of neural circuits across the whole brain be set appropriately in a coordinated fashion. The activation–inactivation continuum – the degree to which the activity of cortical neurons tends to fluctuate synchronously and in phase on timescales of hundreds of milliseconds (*Berger, 1929*; *Steriade et al., 1990*; *Vanderwolf, 2003*) – and pupil dilation – a measure of cognitive load and arousal (*Kahneman and Beatty, 1966*; *Bradley et al., 2008*) – are commonly used to label these large-scale dynamical regimes, often referred to as 'brain states' (*Gervasoni et al., 2004*; *Castro-Alamancos, 2004*; *Poulet and Petersen, 2008*; *Reimer et al., 2014*; *McGinley et al., 2015a*; *Vinck et al., 2015*). What is the relationship between cortical state and behavior? Although cortical desynchronization during wakefulness in rodents was initially linked to movement during exploration (*Vanderwolf, 2003*), desynchronization and movement can be dissociated (*Reimer et al., 2014*; *Vinck et al., 2015*). In fact, it was demonstrated early

that desynchronization can occur under immobility during visual attention (*Winson, 1972*; *Kemp and Kaada, 1975*), suggesting that desynchronization during waking might signal a state where the animal's cognition is oriented toward the environment (*Vanderwolf, 2003*). Such state would presumably be associated with the ability to perform finer perceptual judgments – a hypothesis consistent with many studies showing that the discriminability of neural sensory representations increases monotonically with the level of cortical desynchronization (*Goard and Dan, 2009*; *Marguet and Harris, 2011*; *Pachitariu et al., 2015*; *Beaman et al., 2017*; *Kobak et al., 2019*). Behavioral studies, however, have not generally confirmed this picture. During GO-NOGO sensory detection tasks, performance and arousal (which tends to be associated with desynchronization [*Reimer et al., 2014*; *McGinley et al., 2015a*]) are related, but in a non-monotonic fashion (*McGinley et al., 2015a*) (but see *Neske et al., 2019*). However, GO-NOGO detection tasks are limited in their ability to decouple sensory discrimination and the tendency of the subject to respond, which is relevant since both aspects are expected to be associated with changes in brain state. Thus, potentially different relationships between cortical state and performance could exist in tasks where responsivity and discrimination accuracy can be decoupled. Two-alternative forced-choice (2AFC) discrimination tasks allow a cleaner separation between responsivity and accuracy, but two studies using this approach failed to find a clear link between desynchronization and perceptual accuracy, pointing instead to a role on task engagement, responsivity, and bias (*Waschke et al., 2019* ; *Jacobs et al., 2020*) (but see *Beaman et al., 2017* for effects of desynchronization during the delay period of a delayed comparison task). Thus, existing evidence suggests that the effects of desynchronization on discriminability at the neural and behavioral levels are not fully consistent, raising questions about the functional role of the desynchronized state. Here, we suggest a possible explanation for this discrepancy, by showing that the effect of desynchronization on accuracy during an auditory 2AFC discrimination task depends strongly on the outcome of the previous trial, and is occluded if trial outcome is ignored.

## Results

### Movement, arousal, and temporal fluctuations in baseline activity

In order to investigate the impact of cortical desynchronization on discrimination accuracy, we recorded population activity from the auditory cortex (*Figure 1—figure supplement 1A*) of head-fixed mice while they performed a 2AFC delayed frequency discrimination task (*Figure 1A–C*, Methods). Electrophysiological recordings were made in an acute configuration, targeting the same location in the auditory cortex of each hemisphere for three consecutive days. Neither the number of units, discrimination accuracy, or reaction time (RT) changed significantly across recording sessions in each mouse (Kruskal–Wallis one-way analysis-of-variance-by-ranks test, $p_{\text{units}_{\text{D1}-3}} = 0.23$, $p_{\text{units}_{\text{D4}-6}} = 0.51$, $p_{\text{accuracy}} = 0.09$, $p_{\text{RT}} = 0.32$; *Figure 1—figure supplement 1B–D*). In addition, we monitored pupil size (PupilS) as well as the overall optic flow (OpticF; *Figure 1D*; Methods) of a video recording of the face of the mouse (*Figure 1—figure supplement 1E*), as a proxy for movement signals known to affect synchronization (*Poulet and Petersen, 2008*; *Niell and Stryker, 2010*; *Vanderwolf, 2003*) and cortical activity (*Stringer et al., 2019*; *Musall et al., 2019*; *Salkoff et al., 2020*).

The dynamical regime of baseline spontaneous activity in the auditory cortex in a period of 2 s prior to the presentation of the stimulus was quantified using two statistics: overall firing rate (FR) across the population, and degree of synchronization (Synch). In order to obtain a measure of synchronization as independent as possible of FR, we quantified Synch for each baseline period relative to surrogates of the spike trains from the same period (thus with equal surrogate FR) but shuffled spike times (*Figure 1E, F*, Methods). This measure is normalized, and would take a value of 1 if neurons were statistically uncorrelated and displayed Poisson-like firing. We found that the resulting Synch and FR measures were effectively uncorrelated (*Figure 1F*), to a much larger extent than previously used measures of synchronization, such as the coefficient of variation of the multiunit activity (*Renart et al., 2010*; *Kobak et al., 2019*) (Methods), which displayed negative correlations with baseline FR (*Figure 1G*).

The coordinated fluctuations responsible for Synch are of low frequency, as evident from trial-to-trial comparison of Synch and the power-spectral density of the MUA (*Figure 1H1*; Methods). In particular, strong desynchronization was associated to a suppression of power in the ~4–16 Hz frequency band relative to a Poisson spike train of the same FR (*Figure 1I*). Analysis of the local-field

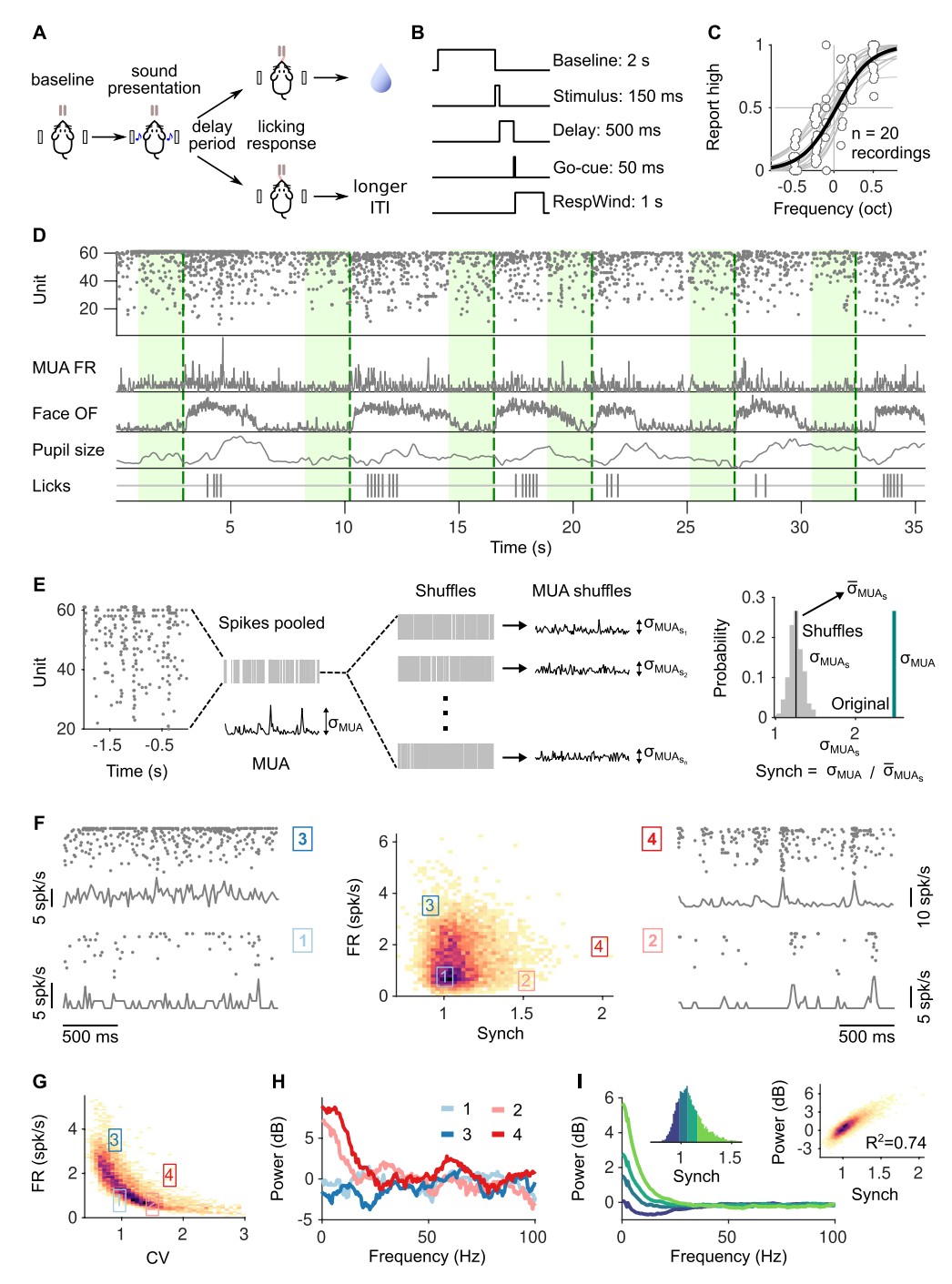

**Figure 1.** Task structure, signals monitored, and quantification of synchrony in baseline activity. (**A**) Task schematic. Head-fixed mice lick at one of two spouts depending on whether the frequency of a pure tone is higher or lower than 14 kHz. (**B**) Temporal sequence of events in a trial. Mice should respond after a delay of 0.5 s. Baseline activity is analyzed in a window of 2 s before the presentation of the sound. (**C**) Discrimination performance. Each dot is the proportion of times a mouse reports high in a given recording session to a given sound. Solid curve is a logistic regression fit. (**D**) Signals monitored. Top to bottom are population raster, multiunit firing rate (MUA FR), mean optic flow of the face (OpticF), size of the pupil (PupilS), and licks. Dashed vertical lines mark stimulus presentation times and green background marks the baseline period we analyze. (**E**) Method for quantifying synchronization. Synch effectively measures the population averaged correlation in the baseline period relative to surrogate data with the same number of spikes but randomly placed in the same period of time (Methods). (**F**) Distribution

*Figure 1 continued on next page*

*Figure 1 continued*

of baseline FR and Synch pooled across all recording sessions. Plots on the sides show rasters and population firing rates for four example baseline periods. (**G**) Identical plot to the one in (**F**)-middle, but where global synchronization is assessed using the coefficient of variation (CV) of the instantaneous population rate (Methods). CV and FR are negatively correlated. (**H**) Power spectrum (Methods) of the four individual example baseline periods in (**F**). (**I**) Average power spectrum of each of the four quantiles of the distribution of Synch across trials. Large values of Synch reflect low-frequency coordinated fluctuations across the population. Inset left: Aggregate distribution of Synch values across recordings. Each quantile corresponds to one of the spectra in panel (**I**). Inset right: Relationship between Synch and average MUA power in the 4–16 Hz range.

The online version of this article includes the following figure supplement(s) for figure 1:

**Figure supplement 1.** Histology, stability of the recordings and behavior over multiple sessions and pupil size analyses.

potential (LFP) was complicated due to the presence of movement artifacts. However, the power of the LFP in the ~4–16 Hz frequency range in baseline periods at the lower end of the OpticF distribution, where movement was largely absent, was significantly correlated with Synch. For instance, for the 2.5% trials with the lowest OpticF, a regression of the $LFP_{4-16Hz}$ power on Synch had an $R^2 = 0.096$ which was highly significant (*t*-test  p < 0.0001).

Although our task is not self-paced and trials arrive in a sequence (Methods), we confirmed that the range of pupil sizes and cortical states that we sample during the pre-stimulus baseline is wide, and depends only weakly on the inter-trial interval (ITI; *Figure 1—figure supplement 1F–J*; Discussion).

Before inspecting the relationship between each of the four signals we analyze (OpticF, PupilS, FR, and Synch) and discrimination performance, we explored the way in which PupilS and OpticF shape baseline neural activity. To do this, we separately regressed FR and Synch on PupilS and OpticF using a linear mixed model with recording session as a random effect (Methods). This analysis revealed FR to be associated to movement and pupil size (*Figure 2A*, top). Surprisingly, Synch did not show a clear association with either predictor (*Figure 2A*, bottom), and a tendency to increase with pupil size, contrary to previous findings (*Reimer et al., 2014*; *Vinck et al., 2015*). Seeking to understand this puzzling result, we inspected more carefully the time series for each of the four baseline signals. This revealed that, in addition to fast trial-by-trial fluctuations, there exist both clear session trends and slow fluctuations spanning many trials, leading to broad auto- and cross-correlations (*Figure 2—figure supplement 1*). These slow components – presumably determined by slow physiological processes (*Okun et al., 2019*) which we do not control – generically lead to correlations between the signals even if the trial-by-trial fluctuations that we are interested in are independent (*Granger and Newbold, 1974*; *Amarasingham et al., 2012*; *Elber-Dorozko and Loewenstein, 2018*; *Harris, 2020*), which can lead to false positive inferences. This is because any two randomly fluctuating variables will generally be empirically correlated – even if generated independently – unless the number of independent samples from each is sufficiently large, as any measure of dependency has an upward bias for limited numbers of samples (*Treves and Panzeri, 1995*). For time series, the effective number of independent samples is their duration in units of the timescale of their temporal correlations. Thus, if the temporal correlations of our signals are long lived, and comparable to the session length, the number of effectively independent samples will be low, and any two signals will in general be empirically correlated. To address this problem and try to minimize the probability of making false positive inferences, we sought to remove the slow fluctuations in our signals. To do this, we first fit a linear regression model to each signal, trying to predict its value in each trial as a linear combination of: its own value and the value of the other signals and trial outcomes in the previous 10 trials, and the session trend (Methods; *Figure 2—figure supplement 2A*). Then we defined the 'innovation' associated to each signal (which we denote with the subscript I, e.g., FR₁) as the difference between the value of the signal in one trial, and its predicted value (*Kailath, 1968*), that is, as the residual of this linear model.

Different signals could be predicted by past information to different extents, with Synch and PupilS being the least and most predictable, respectively ($r^2_{Synch} = 0.22 \pm 0.04$ and $r^2_{PupilS} = 0.55 \pm 0.13$; median ± median absolute deviation [MAD] across recordings; *Figure 2—figure supplement 2B, C*). Innovations, on the other hand, displayed effectively 'white' auto- and cross-correlations (*Figure 2C*). Thus, any associations revealed using innovations as regressors will not be caused by random empirical

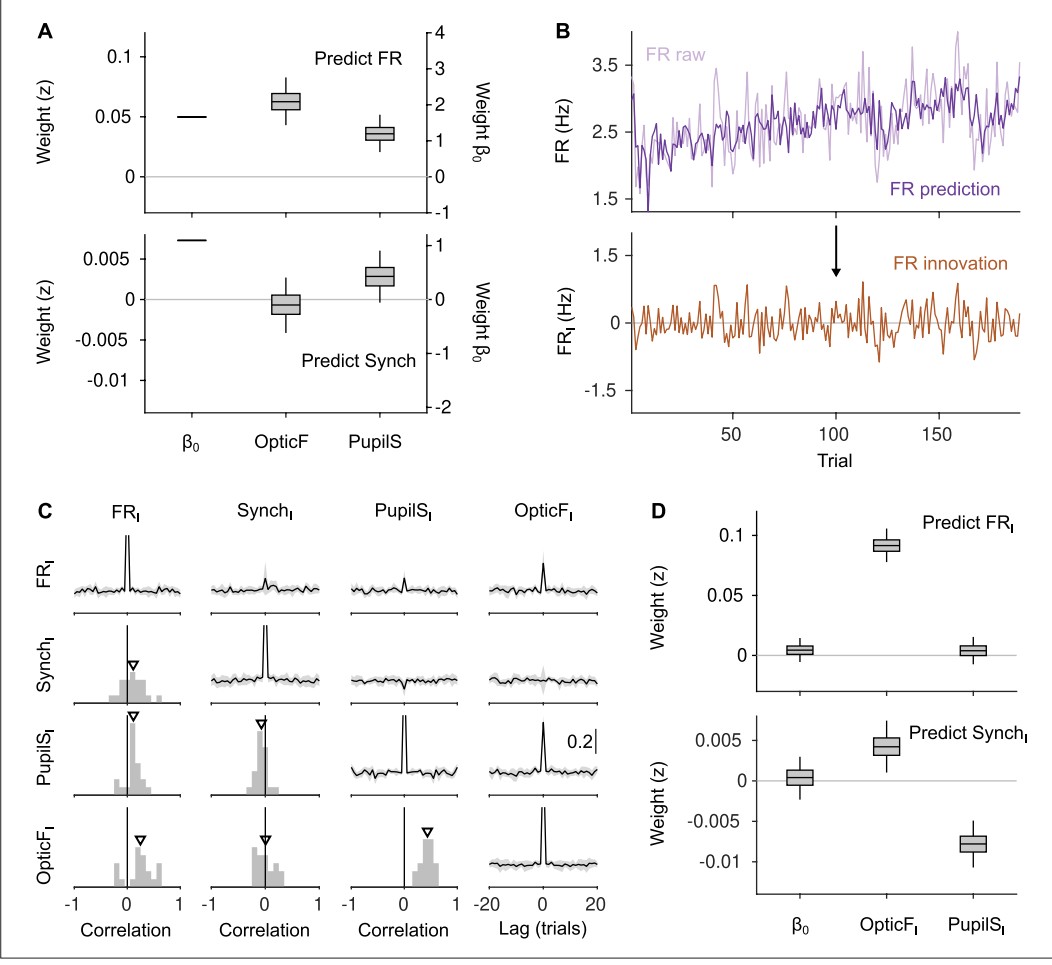

**Figure 2.** Innovations clarify the effect of movement and pupil size on cortical state fluctuations. (**A**) Linear mixed model regression (Methods) of firing rate (FR; top) and Synch (bottom) on movement and pupil size. Graphs show values of regression coefficients. Box plots here and elsewhere represent median, interquartile range and 95% confidence interval (CI) on the bootstrap distribution of the corresponding parameter (Methods). Offset can be read from the right y-axis. (**B**) Example of the process of calculating innovations for the baseline FR of one recording session. Top, raw data and prediction of the raw data (*Figure 2—figure supplement 2*; Methods). The innovation FR$_I$ (bottom) is the difference (prediction residual) between the two traces in the top. (**C**) Correlation between OpticF, PupilS, FR, and Synch innovations. Diagonal and above, cross-correlations between each of the four signals (black, median across recordings; gray, median absolute deviation [MAD]). Below diagonal. For each pair of innovations, histogram across recordings of their instantaneous correlation. Triangles mark the median across recordings. (**D**) Identical analysis as panel (**A**) but using innovations instead of the raw signals.

The online version of this article includes the following figure supplement(s) for figure 2:

**Figure supplement 1.** Slow trends of baseline signals during the session.

**Figure supplement 2.** Constructing innovations by cross-whitening.

associations between the slow components of the signals (*Granger and Newbold, 1974*; *Amarasingham et al., 2012*; *Elber-Dorozko and Loewenstein, 2018*; *Harris, 2020*).

When the analysis in *Figure 2A* was repeated using innovations, a different picture emerged. Although FR$_I$ is positively correlated with both OpticF$_I$ and PupilS$_I$ (*Figure 2C*), the correlation with PupilS$_I$ is explained away by the positive correlation between OpticF$_I$ and PupilS$_I$ themselves, revealing a clear positive association only between movement and FR innovations during the baseline (p < 0.0002, bootstrap quantile method [*Efron and Tibshirani, 1994*], from now on referred to as 'bootstrap'; Methods). Synch$_I$ is much more weakly correlated with both OpticF$_I$ and PupilS$_I$ (*Figure 2C, D*). Nevertheless, the analysis revealed a positive association between pupil size and desynchronization (p < 0.0002, bootstrap) – consistent with previous studies (*Reimer et al., 2014*; *Vinck et al., 2015*) – as

well as a rather small but significant (p = 0.012, bootstrap) positive association between movement and synchronization (*Figure 2D*). For the rest of our study, we seek to explain choice behavior in terms of innovations to characterize trial-by-trial relationships between discrimination accuracy and brain state (although we also use the raw signals as regressors in control analyses).

## Outcome-dependent effect of desynchronization on choice accuracy

We used a generalized linear mixed model (GLMM; Methods) to explain whether each trial was correct or an error based on the strength of sensory evidence (Stim) and the four innovations during the baseline preceding that trial. We sought to predict whether a choice was correct rather than the choice itself (left versus right) so that the potential effect of innovations would represent a main effect in the model, rather than an interaction with the stimulus (but see *Figure 3—figure supplement 1B*). This analysis only considers valid trials (Methods) where the mice made a choice within the response window, and thus quantifies the effect of brain state on discrimination accuracy regardless of unspecific response tendencies. In order to be able to explain within-session trends, we always include a regressor coding the trial number within the session (TrN). Finally, to model possible sequential dependencies in choice accuracy, we also included a regressor with the outcome (correct/error) of the previous trial (pCorr; only valid previous trials were considered). The analysis revealed a positive association between TrN and accuracy (*Figure 3A*; p = 0.005, bootstrap) – reflecting the fact that mice tend to become more accurate throughout the session – but none of the four baseline predictors had an association with accuracy, consistent with a recent study (*Jacobs et al., 2020*; *Figure 3A*; a table in *Supplementary file 1* lists the complete results of all GLMM fits in the main text). However, the coefficient measuring the effect of the outcome of the previous trial was negative (p = 0.006, bootstrap), suggesting that mice tended to be more accurate after errors (*Figure 3A*). Indeed, across sessions, accuracy was larger after an error (*Figure 3B*; p = 0.021, signrank test, Methods). It is well known that errors have an effect on the RT of the subsequent trial (*Rabbitt, 1966*; *Laming, 1979*; *Danielmeier and Ullsperger, 2011*), and, although less consistently, accuracy enhancements after an error have also been observed (*Laming, 1979*; *Marco-Pallarés et al., 2008*; *Danielmeier and Ullsperger, 2011*). Given that errors have an impact on task performance, we reasoned that they might modulate the role of spontaneous cortical fluctuations on choice. To test this hypothesis, we performed our analysis separately after correct and error trials. The results revealed that, while pupil size and movement still had no association with accuracy for either outcome separately (*Figure 3C, E*), the effect of baseline neural activity on choice accuracy was indeed outcome dependent (*Figure 3C–F*). After errors, both FR and Synch innovations in the baseline period explain accuracy (*Figure 3C*; p = 0.0056 and p = 0.0124 for $FR_I$ and $Synch_I$, respectively; bootstrap).

Mice made more accurate decisions when the baseline activity was higher and more desynchronized, a state we refer to as 'favorable' for accuracy after an error. In contrast, baseline activity had no clear association to accuracy after correct trials (*Figure 3E*; p = 0.64 and p = 0.22 for $FR_I$ and $Synch_I$, respectively; bootstrap), despite the fact that the GLMM for after-correct choices had approximately three times as many trials (which is reflected on the smaller magnitude of the confidence intervals [CIs] for this model; *Figure 3E*). Although this makes it difficult to define a 'favorable' state for accuracy after correct trials, the median value of the coefficients for both $FR_I$ and $Synch_I$ in *Figure 3E* is positive, suggesting that, if anything, more accurate choices after a correct trial were preceded by more synchronized (and stronger) baseline activity. The lack of effect of baseline activity on accuracy unconditional on outcome (*Figure 3A*) is explained partly by the tendency of baseline fluctuations preceding a correct choice to have different signs (relative to the mean) after correct and error trials and by the fact that most trials (77%) are correct.

To assess together the effect of baseline FR and Synch innovations on accuracy, we created a single predictor for each baseline period whose value was equal to the projection of the (*z*-scored) two-dimensional pair ($Synch_I$, $FR_I$) onto a line of slope −45 deg on this plane (after errors), or 45 deg after corrects (Methods). This single predictor takes large positive values when both FR and Synch are 'favorable' for accuracy for each separate outcome. After errors, the combined effect of FR and Synch was 28% stronger than that of either of them separately and highly significant (*Figure 3C*, rightmost coefficient; p = 0.0006, bootstrap), but it was still not significant after correct choices (*Figure 3E*, rightmost coefficient; p = 0.2, bootstrap). To more directly quantify the effect of baseline neural activity on accuracy, we also computed aggregate psychometric functions for trials where the state of

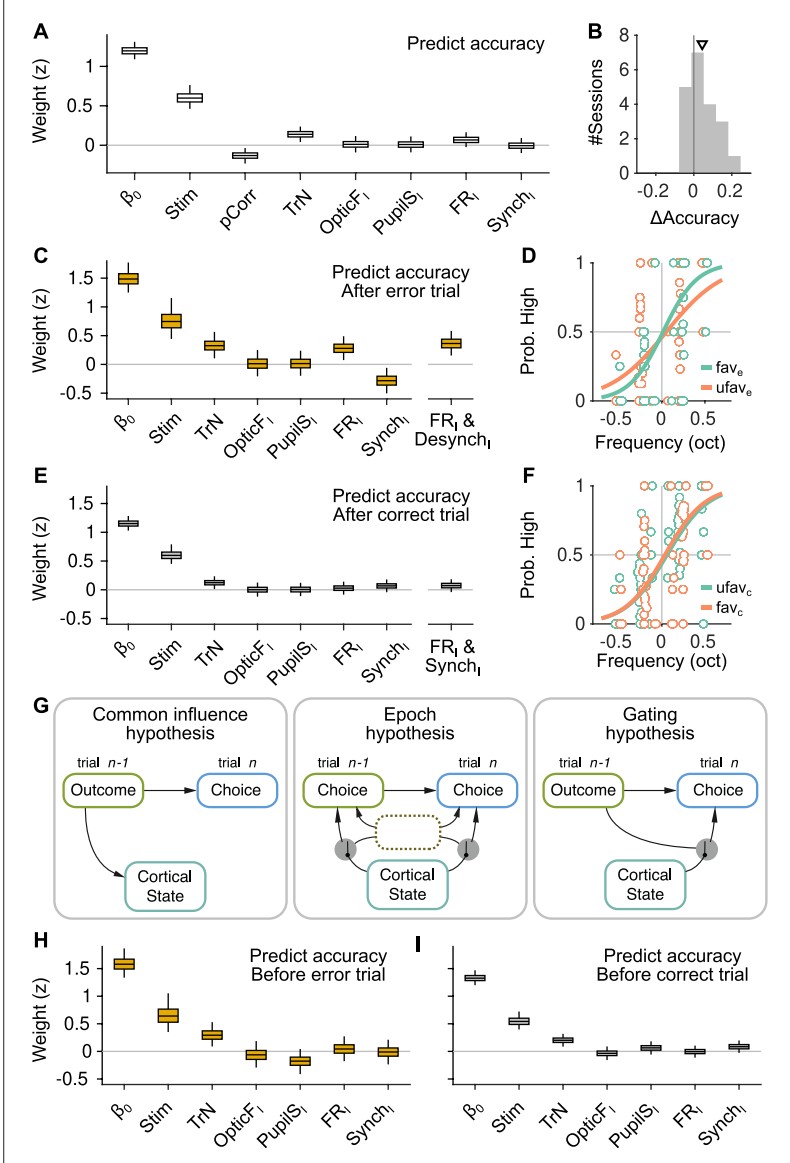

**Figure 3.** The effect of spontaneous state fluctuations on accuracy is outcome dependent. (**A**) Coefficients of a generalized linear mixed model (GLMM) fit to the mice's choice accuracy in valid trials. Accuracy is affected by the strength of evidence, the point during the session and the outcome of the previous trial, but none of the four signals computed during the baseline explain accuracy. (**B**) Mean difference in accuracy after errors minus after corrects in each of the recording sessions. Triangle, median across sessions. (**C**) GLMM fit to accuracy computed separately after error trials. On the right, we show the distribution of a single coefficient capturing trial-to-trial fluctuations in desynchronization and firing rate simultaneously (see text). (**D**) Psychometric function (logistic fit, Methods) of aggregate data across sessions separately for trials with favorable ($\text{Synch}_I(z) < 0$ and $\text{FR}_I(z) > 0$) and unfavorable ($\text{Synch}_I(z) > 0$ and $\text{FR}_I(z) < 0$) baseline states after a error trials. (**E, F**) Same as (**C, D**) but for choices after a correct trial. Note that, based on the results in (**E**), the favorable state after a correct trial is $\text{Synch}_I(z) > 0$ and $\text{FR}_I(z) > 0$. (**G**) Schematic illustration of possible relationships between outcome, baseline cortical state and accuracy. Left, the association between state and accuracy is spurious and results from a common effect of response outcome on these two variables. Middle, epoch hypothesis (see text). An unmeasured variable with a timescale of several trials mediates both the effect of state on accuracy and the prevalence of errors. Right, response outcome gates the effect of state fluctuations (errors open the gate) on choice accuracy. (**H, I**) Same as (**C, E**) but conditioned on the outcome of the next, rather than the previous trial.

The online version of this article includes the following figure supplement(s) for figure 3:

**Figure supplement 1.** Robustness of the association between brain state and accuracy.

*Figure 3 continued on next page*

*Figure 3 continued*

**Figure supplement 2.** Generalized linear mixed model (GLMM) analysis including quadratic terms and differentially for superficial and deep recording shanks.

**Figure supplement 3.** Robustness of the results on the effects of cortical state on accuracy.

**Figure supplement 4.** Lack of association between slow cortical state fluctuations and accuracy.

**Figure supplement 5.** Behavioral predictions including slow trends.

**Figure supplement 6.** Outcome dependence of the effect of cortical state on stimulus and choice discriminability from evoked responses.

the baseline was favorable or unfavorable, separately after correct and error trials. The slope of the psychometric function was 68% larger in a favorable baseline ($\text{Synch}_I(z) < 0$ and $\text{FR}_I(z) > 0$) after errors (*Figure 3D*; p = 0.04, permutation test, Methods). There was no visible effect of a favorable state after a correct trial (during which the cortex was more synchronized) on the aggregate psychometric function (*Figure 3F*, p = 0.88, permutation test).

We tested the robustness of this finding in various ways. The outcome dependence of the effect of baseline fluctuations on accuracy was qualitatively similar when assessed using using parametric methods for the calculation of CIs (*Figure 3—figure supplement 1A*, Methods). Results were also consistent when predicting trial-by-trial choice (as opposed to accuracy; *Figure 3—figure supplement 1B*, Methods). Choice predictions allowed us to test whether cortical state has an effect of choice bias, as some authors have observed previously (*Waschke et al., 2019*). In our dataset, cortical state was only predictive as an interaction term (i.e., it had an effect on sensitivity, not criterion), and only after errors (*Figure 3—figure supplement 1B*). Considering recording sessions as random effects nested within mice also gave similar results (*Figure 3—figure supplement 1C*).

In GO-NOGO detection tasks, the effect of arousal on accuracy is sometimes non-monotonic (*McGinley et al., 2015a*). To test for the possibility of non-monotonicity in the relationship between $\text{FR}_I$ and $\text{Synch}_I$ and accuracy we included quadratic terms in our predictive models (*Figure 3—figure supplement 2A*). The presence of quadratic terms did not alter the finding in *Figure 3C, E*. The only significant quadratic coefficient was the one for $\text{FR}_I$ after errors, revealing a monotonic but accelerating dependence of accuracy on $\text{FR}_I$ (*Figure 3—figure supplement 2B*). Our probe insertion strategy places the shanks of the silicon probe in a coronal plane, with each shank roughly parallel to the cortical layers (*Figure 1—figure supplement 1A*). We used this arrangement to assess whether the results in *Figure 3C, E* held when defining measures of cortical state ($\text{FR}_I$ and $\text{Synch}_I$) using neurons recorded in the three most superficial (deep) shanks, which will largely be located in the most superficial (deep) cortical layers. Using these putatively superficial or deep neural populations (fraction of superficial neurons relative to the total 0.55 ± 0.08, median ± MAD) produced a similar general pattern of results as the aggregate result in *Figure 3C, E* (*Figure 3—figure supplement 2C, D*), although the magnitude of the coefficients associated to the $\text{FR}_I$ and $\text{Synch}_I$ predictors was weaker, presumably because the estimation of cortical state suffers from using neural populations of approximately half the size. Finally, in terms of the time window used to define the baseline period, the predictive power of $\text{FR}_I$ and $\text{Synch}_I$ on accuracy degraded gradually if the window became too small (0.5 or 1 s instead of 2 s) or moved away from stimulus presentation ([−4 −2] s instead of [−2 0] s relative to stimulus onset), suggesting that the baseline state should be defined and can change in a timescale of seconds (*Figure 3—figure supplement 3*).

What exactly do the results in *Figure 3A–F* imply for the relationship between spontaneous baseline activity and choice? An explanation of these results as a spurious correlation caused by the joint influence of the outcome of the previous trial on accuracy and on baseline activity in the current trial (*Figure 3G*, left) can be ruled out, since the outcome of the previous trial is fixed in the analyses of *Figure 3C, E*. Rather, our results suggest that errors gate, or enable, the influence of spontaneous fluctuations on choice (*Figure 3G*, right). However, it is still possible the gating is not performed by errors per se, but rather by some other quantity that tends to covary in time with errors. In other words, there might be epochs within the session during which spontaneous cortical fluctuations have an effect on accuracy and during which errors are more frequent (*Figure 3G*, middle). We refer to this as the 'epoch hypothesis'. The epoch hypothesis can be tested under the assumption that the epochs last a few trials, in which case the relationship between baseline activity and accuracy should

be approximately symmetric around the time of an error. To test if this is the case, we repeated the analysis in *Figure 3C, E*, but instead of conditioning on the outcome of the previous trial, we conditioned on the outcome of the next trial. If the epoch hypothesis is true, we would expect for $FR_I$ and $Synch_I$ to explain accuracy in a trial when the next trial is an error, just like in *Figure 3C*. In contrast, we found that baseline fluctuations have no predictive power on the accuracy of a trial regardless of the outcome of the next trial (*Figure 3H1*). If trial $n + 1$ is correct, the influence of $Synch_I$ and $FR_I$ on the accuracy in trials $n$ is similar to that observed if trial $n - 1$ is correct: not significantly different from zero but with a tendency toward higher accuracy when the baseline is more synchronized (*Figure 3E, I* ). In contrast, baseline activity is clearly predictive of choice accuracy in trial $n$ only if an error takes place in trial $n - 1$, but not on trial $n + 1$. These results are inconsistent with the idea that errors mark epochs of high correlation between cortical fluctuations and accuracy, and support instead the hypothesis that this correlation is triggered by the errors themselves (*Figure 3G*, right).

By construction, slow fluctuations in the baseline signals do not contribute to the effects in *Figure 3*, but the four 'raw' baseline signals do display such slow fluctuations (*Figure 2—figure supplement 1*). We investigated if slow fluctuations were associated to discrimination accuracy in two different ways. First, we smoothed, linearly detrended and *z*-scored the raw baseline FR and Synch time series and the corresponding accuracy in those trials (*Figure 3—figure supplement 4A*, Methods), and computed their cross-correlation. We observed no correlations between Synch and accuracy (*Figure 3—figure supplement 4B*; p = 0.94, signrank test) and a trend toward epochs of high performance to precede epochs of low baseline FR (*Figure 3—figure supplement 4B*; p = 0.1, signrank test). Second, we directly ran our predictive models conditioned on the outcome of the previous trial (*Figure 3C, E*) but using the raw signals, instead of their innovations. The predictive power of the regressors corresponding to the raw signals are qualitatively similar to those corresponding to their innovations (*Figure 3—figure supplement 5A,B and E,F*; *Figure 3C, E*). This suggests that, in our experiments, slow trends in cortical state, pupil size or facial movement are not associated with discrimination accuracy in a way that is consistent across recording sessions.

Finally, we studied the sound-evoked activity of the recorded neurons (in a [0 150] ms window relative to stimulus onset) to assess whether baseline activity and previous trial outcome shaped the representation of sounds by neurons in the auditory cortex or its relationship to choice (*Figure 3—figure supplement 6A*). In order to be able to aggregate data across sessions, we first defined a 'sound axis' in each recording separately by predicting the sound category (whether a sound required a lick to either of the two spouts) using cross-validated regularized logistic regression (Methods). Because the overall performance in the task is above chance, stimulus and choice are correlated, so we calculated the sound axis separately for each choice. The same exact procedure was used to define a 'choice axis' separately for each stimulus. We then obtained a scalar, single-trial measure of stimulus or choice discriminability by projecting the high-dimensional evoked activity in that trial on the corresponding axis (*Figure 3—figure supplement 6B*). Next, we ran a GLMM (using recording session as a random effect) to predict the stimulus category or choice on aggregate across experiments, separately after each outcome. An effect of baseline state on sound-evoked stimulus or choice discriminability can be detected as a non-zero interaction term between the stimulus or choice projection regressors and $FR_I$ or $Synch_I$ (*Figure 3—figure supplement 6C*).

Regarding stimulus discriminability, there was a clear main effect of the stimulus projection regardless of outcome (*Figure 3—figure supplement 6D*), suggesting that the stimulus category could be decoded from the evoked responses of the recorded neurons. None of the interaction terms were significantly different from zero after correct trials. After errors, the interaction term between between $FR_I$ and the stimulus projection was significantly positive (p = 0.002; conditional mean squared error of prediction [CMSEP] method; Methods). The median of the $Synch_I$ interaction was negative, but it was not significant (p = 0.4; 95% CI = [−0.32,0.13]). Thus, consistent with the results in *Figure 3C, E*, the favorable state for accuracy after errors is also associated to better stimulus discriminability in the auditory cortex, although, at the level of $Synch_I$, the effect does not reach significance (*Figure 3—figure supplement 6D*).

The same type of analysis revealed that there was no main effect of the choice projection regardless of outcome (*Figure 3—figure supplement 6E*), indicating that choice-related signals in the evoked activity of the recorded neurons in our dataset were too weak to be detected. The fact that choice-related signals in sensory areas are typically small (*Crapse and Basso, 2015*), and the fact that

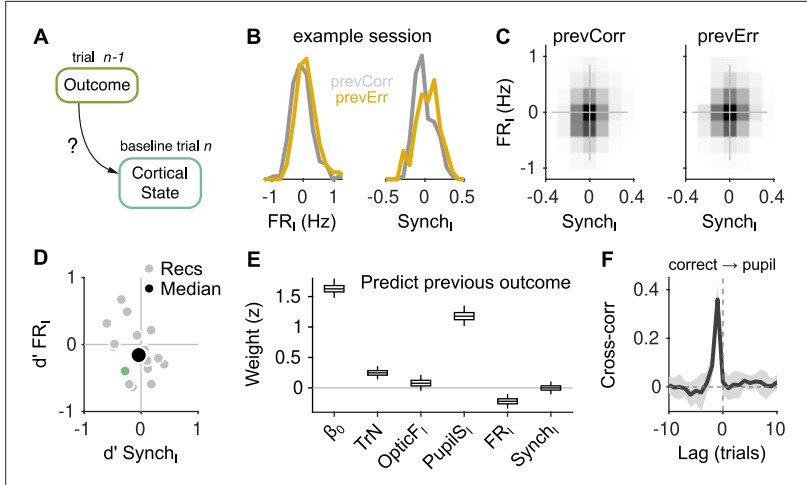

**Figure 4.** Effect of outcome on baseline activity. (**A**) Schematic illustration of the question addressed in this figure. (**B**) Distribution of $FR_I$ (left) and $Synch_I$ (right) after each of the two outcomes for an example session. (**C**) Joint histogram of $FR_I$ and $Synch_I$ on aggregate across recordings after a correct (left) and after an error (right) trial. (**D**) Discriminability index $d'$ between the distributions of $FR_I$ and $Synch_I$ (such as those in (**B**)) after each of the two outcomes. Each gray dot corresponds to one recording, the colored dot is the example recording in (**B**), and the large black circle is the median. (**E**) Coefficients of a generalized linear mixed model (GLMM) fit to the outcome (correct or error) of the mice's choices on trial $n - 1$ using as regressors TrN and innovations from the baseline of trial $n$. (**F**) Cross-correlation function between the raw outcome and PupilS time series (Methods). Black is the median across recordings, gray is the median absolute deviation (MAD). Throughout this figure, innovations were modified so as to exclude previous outcomes in the calculations of the residuals (Methods).

the choice axis needs to be estimated experiment by experiment using small nubers of trials (specially after errors [median of 13–14 errors per experiment per stimulus category], which is were **Figure 3C** suggests choice-related signals might be present), could explain this result.

## Cortical fluctuations are only weakly affected by trial outcome

We next sought to understand whether the selective influence of baseline activity on choice after errors (**Figure 3**) is due to a particular pattern of cortical state fluctuations that is only evident after the mouse makes a mistake. For instance, it is possible that desynchronization is always conducive to better performance, but that sufficient levels of desynchronization are only attained after errors. We explored this question by quantifying the extent to which trial outcome shapes cortical state fluctuations (**Figure 4A**). To accomplish this, it is necessary first to modify the way we calculate the baseline signal's innovations, as they are defined to be automatically uncorrelated with the outcome of the previous trial (Methods, **Figure 2—figure supplement 2**). Thus, we simply excluded previous-trial outcome from the linear model used to predict the baseline signals in each trial, before calculating the residuals (Methods).

The values of $Synch_I$ and $FR_I$ observed after an error or a correct trial largely overlap (**Figure 4B**), and the joint distribution of $Synch_I$ and $FR_I$ across recordings are very similar (**Figure 4C**). We first quantified these effects calculating the signed discriminability index $d'$ (correct minus errors) of the distributions of $Synch_I$ and $FR_I$ for each recording. Across recordings, neither of these two measures were significantly different from zero ($p = 0.26$ and $p = 0.13$ for $Synch_I$ and $FR_I$, respectively; sign-rank test). As an alternative, more sensitive approach to understand which features of the baseline contained information about the outcome of the previous trial, we used a GLMM to decode whether the outcome of trial $n - 1$ was correct, using as regressors the four innovations in the baseline of trial $n$, as well as the session trend TrN. Previous-trial outcome is best explained by the $PupilS_I$ in the subsequent baseline (**Figure 4E**). This is intuitively clear, as correct trials are followed by licking, which is associated to pupil dilation (**Cazettes et al., 2021**), a relationship that becomes obvious when plotting the cross-correlation function between the accuracy and PupilS time series (**Figure 4F**). In addition to the pupil size, $FR_I$ is also affected by the outcome of the previous trial, being smaller than average after correct trials (consistent with the small negative median value of $d'_{FR_I}$ in **Figure 4D** and with the

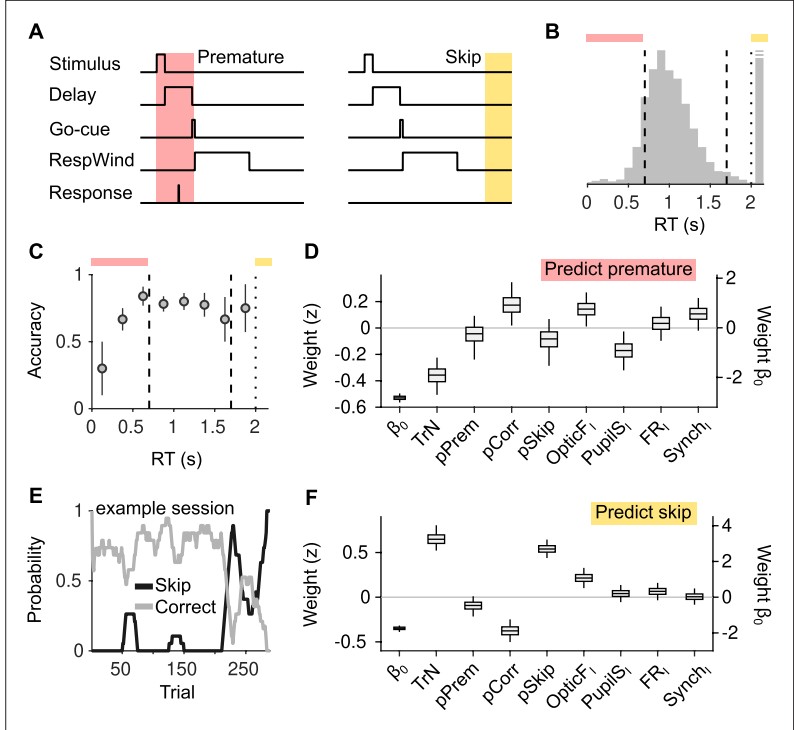

**Figure 5.** Effect of cortical state fluctuations on premature responding and engagement. (**A**) Definition of premature responses and skips. (**B**) Aggregate across sessions of the distribution of reaction times (RTs) in our task. Dashed lines indicate the response window in which a correct response was rewarded (valid trials). Trials where a response is not produced before the dotted line are defined as skips. Top, colors used to signal each trial type in (**B**). (**C**) Accuracy (median ± median absolute deviation [MAD] across recordings) conditional on RT. (**D**) Coefficients of a generalized linear mixed model (GLMM) fit to explain whether a given trial is premature or valid. Magnitude of the offset ($\beta_0$) should be read of from the right y-axis. (**E**) Probability of not responding to the stimulus (skip) in an example session. Skips tend to occur in bouts and are more frequent toward the end of the session. (**F**) Same as (**D**) but for a GLMM aimed at explaining if a particular trial is a skip or valid.

The online version of this article includes the following figure supplement(s) for figure 5:

**Figure supplement 1.** Explaining reaction time (RT) in valid trials.

negative trend in *Figure 3—figure supplement 4B*). $Synch_I$ could not be used to predict the outcome of the previous trial. Overall, these results are not consistent with the effects in *Figure 3* being due to the presence of unique values of FR and Synch exclusively after errors. Errors do increase the FR in the next baseline period, but FR distributions after the two outcomes are largely overlapping. In addition, and somewhat unexpectedly, trial outcome has no effect at all on baseline synchrony.

## Effect of spontaneous fluctuations on measures of responsivity

Arousal and desynchronization have been shown to modulate measures of responsivity (*McGinley et al., 2015a*; *Waschke et al., 2019*; *Jacobs et al., 2020*). There are two different facets to responsivity in a discrimination task. One relates to the tendency of the subject to respond at all to a presented stimulus, which can be taken as a measure of task engagement. The other is RT, the time (since stimulus onset) it takes for the subject to respond. In a delayed response task like ours, there is additionally the possibility for mice to respond prematurely, failing to wait for the go signal at the end of the delay period (*Figure 5A*). In our task, most trials were valid (either correct or errors, 70%, Methods), but there were also premature trials (7%) and 'skips' where the mice did not respond (23%; *Figure 5B*).

Choice accuracy varied as a function of RT (*Figure 5C*). Very premature responses where most inaccurate. Accuracy tended to increase with RT for premature responses during the delay period, and then remained approximately constant within the valid response window and beyond. These results suggest that premature and valid responses might be differentially regulated. We explored

this possibility by trying to explain whether a trial would be premature or valid using a GLMM. Unlike *Figures 3 and 4*, which only deal with transitions between valid trials, here the previous trial could be either valid, premature, or a skip, and we thus included corresponding regressors in the GLMM (Methods). The most reliable predictor of a premature trial was TrN (*Figure 5D*; p < 0.0002, bootstrap), signaling a decreasing tendency to respond prematurely as the session progresses, paralleling changes in motivational state (*Berditchevskaia et al., 2016*). Everything else kept equal, premature trials also happened more frequently after correct trials (p = 0.03, bootstrap), in the presence of movement in the baseline (p = 0.03, bootstrap), and when cortical activity was more synchronized, although this last effect did not reach significance (p = 0.09, bootstrap). Interestingly, baseline periods with contracted pupil were predictive of premature responses (p = 0.02, bootstrap). Although this finding might seem at odds with previously reported associations between states of dilated pupil and impulsivity (*McGinley et al., 2015a*; *Jacobs et al., 2020*), a large body of work has linked pupil dilation with the ability to exercise inhibitory control (*Wang et al., 2015*; *van der Wel and van Steenbergen, 2018*), which is needed in order to avoid responding prematurely (Discussion).

Although RT did not primarily reflect decision time and was instead constrained by the delay period of the task (*Figures 1A, B and 5A*), we nevertheless used a similar approach to explore a possible effect of cortical desynchronization on RT. Only movement innovations were positively associated with RT (p = 0.01, bootstrap; *Figure 5—figure supplement 1*). Somewhat surprisingly, we observed evidence of post-error slowing (*Rabbitt, 1966*; *Laming, 1979*; *Danielmeier and Ullsperger, 2011*), suggesting that the connection between errors and subsequent RT is so strong that it survives the constraints in RT imposed by a delayed response task. Just as with our models of choice accuracy, the predictive power of all regressors in models of RT was qualitatively similar using the raw signals or their innovations (*Figure 5—figure supplement 1B, C*).

Finally, we examined engagement. As commonly observed, mice underwent periods of disengagement during behavioral sessions (*Ashwood et al., 2022*; *Jacobs et al., 2020*), defined as bouts of consecutive trials during which the mice did not respond to the stimuli ('skips', *Figure 5A,E*). We attempted to predict whether a trial would be a skip or valid using identical regressors as for premature responses. Opposite to premature trials, skips were more frequent at the end of the session (*Figure 5F*, p < 0.0002, bootstrap), and were, everything else kept equal, more frequent after skips and less likely after correct trials. Of the four signals in the baseline, only OpticF$_I$ had a positive significant association with skips (*Figure 5F*; p = 0.0006, bootstrap), suggesting that mice are more likely to perform facial movements while they are distracted from the task. FR and Synch innovations had no explanatory power for skips (p = 0.19 and p = 0.90 for FR$_I$ and Synch$_I$, respectively; bootstrap). Thus, cortical desynchronization innovations had no association with engagement for our mice (*Figure 5F*). As for choice accuracy, we repeated our analysis using the raw baseline signals, instead of their innovations, in predictive models of premature responses or Skips. Again, we found the results were very similar with or without innovations (*Figure 3—figure supplement 5*), suggesting no consistent associations across recording sessions between the slow components of the baseline signals and behavior.

## Discussion

Our main finding is that the effect of spontaneous cortical fluctuations on perceptual accuracy is only evident after errors, with mice making more accurate choices after errors when baseline activity in the auditory cortex was higher and more desynchronized (*Figure 3C–F*). This outcome dependence could not be explained through the existence of epochs where cortical fluctuations are linked to accuracy and where errors are simultaneously more prevalent (*Figure 3H*), nor through the presence of a particular baseline state favorable for accuracy found only after errors (*Figure 4*). Instead, errors appear to permit baseline fluctuations to become associated with choice accuracy, consistent with a gating role. Discrimination accuracy was not associated to pupil dilation or facial movement during the baseline, but these two signals did show associations with measures of responsivity. Pupil dilation predicted the ability of the mice to withhold responding during the delay period, an ability which tended to also be associated with desynchronization (although not significantly, *Figure 5D*). Facial movement clearly predicted whether the mice would disengage in a particular trial (*Figure 5F*) and also, to a smaller extent, premature responding (*Figure 5D*), whereas baseline neural activity did not predict engagement (*Figure 5F*).

A possible limitation of our study is that our recording strategy did not allow us to quantify the spatial resolution to the brain–behavior relationships we describe. We recorded from the same mice during several consecutive days using different penetrations (targeted to the same location in the auditory cortex; Methods), which precluded the histological reconstruction of the tracts left by the silicon probe in most recording sessions. However, previous work suggests that, at least in rodents, dynamical states associated with specific patterns of choice are shared across large regions of the cortex (*Jacobs et al., 2020*).

Another possible concern is that our task is not self-paced, and each trial is followed by another after a period of 3–12 s, depending on outcome (Methods). The corresponding sequence of stimuli might 'over arouse' the mice, that is, preventing us from sampling a wide range of cortical states and, in particular, to sample periods of low arousal. To investigate whether this is the case, we evaluated how the distributions of pupil size, FR and Synch depend on the ITI. For all ITIs, the distribution of the pupil had at least 25% of its probability mass for pupil sizes lower than 20% of its minimum within-session value (*Figure 1—figure supplement 1H*), suggesting that constricted pupils are broadly sampled in our task regardless of the ITI. Furthermore, the range of the pupil size distribution was constant for ITIs longer than 9 s (quantile linear regression of 2.5 and 97.5 percentiles of the pupil against ITI; 95% $CI_{2.5}$ = [−0.32 0.54]; 95% $CI_{97.5}$ = [−0.03 1.34]; bootstrap), implying that the pupil size distribution reaches its steady state at ITIs of approximately 10 s and that longer ITIs will not lead to further changes. A similar analysis on the two measures of cortical state revealed that the distribution of both FR and Synch did not change as a function of ITI (*Figure 1—figure supplement 1I*). Finally, we compared the shape of the pupil size distribution (identically pre-processed and normalized) in our task and in a foraging task we have described previously (*Cazettes et al., 2021*), where trials are self-paced and mice run in a treadmill. Compared to our task, the pupil distribution in the treadmill task has more mass at the dilated end, coming from periods of locomotion. However, this is at the expense of less mass at intermediate pupil sizes: the constricted end of the pupil distribution is completely overlapping across the two tasks (*Figure 1—figure supplement 1J*). In summary, although the extent to which our findings depend on the specific features of our task is an empirical question that will need to be addressed in future studies, these analyses suggest that the parameters of our task do not particularly restrict the range of cortical states or pupil-linked arousal that we sample.

There is renewed awareness (*Elber-Dorozko and Loewenstein, 2018*; *Harris, 2020*) that observed covariations between neural activity and behavior might be spurious, in the sense of reflecting 'small sample' biases when slow trends are present in predictors and prediction targets (*Granger and Newbold, 1974*). Such slow trends are ubiquitous (since many physiological processes vary slowly) and, indeed, the physiological and behavioral signals we analyzed all displayed slow trends of variation across the recording session (*Figures 3–5*, *Figure 2—figure supplement 1A*) as well as auto- and cross-correlations spanning several trials (*Figure 2—figure supplement 1B*). We attempted to avoid this problem by regressing the behavior of the mice on pre-processed versions of the baseline signals that were cross-whitened (Methods, *Figure 2—figure supplement 2*) in such a way that they did not display any temporal correlations (with themselves or with each other) across trials (*Figure 2B, C*), which we called innovations. Results obtained using innovations reflect associations that are taking place between the fast, trial-by-trial components of the baseline signals and behavior. For completeness, however, we also conducted our analyses using the raw baseline signals (*Figure 3—figure supplement 5*, *Figure 5—figure supplement 1*). As long as the session trend was included as a regressor in the predictive models, these two approaches gave similar results, suggesting that all the brain–behavior relationships in our dataset which are systematic across recording sessions, take place between the fast components of the measured signals. Considering predictive models with either temporally uncorrelated or the raw regressors can provide information about the timescale at which the observed brain–behavior relationships are taking place. In particular, the difference between the magnitude of the fixed component of a given regression coefficient without or with innovations is a measure of the systematic association between the slow components of the corresponding regressor and the output across sessions. In our dataset, these slow associations are marginal.

What can be the mechanistic implementation of the outcome-dependent gating of cortical state fluctuations on choice? The fact that the coupling takes place only after errors suggests the involvement of anterior cingulate and medial frontal brain areas, which are associated with performance monitoring and cognitive control (*Botvinick et al., 2001*; *Ridderinkhof et al., 2004*; *Ullsperger*

*et al., 2014*). The relationship between errors and cognitive control is well established (*Botvinick et al., 2001*; *Ullsperger et al., 2014*), and probably arises both through the effect of errors on motivation (*Botvinick and Braver, 2015*) (errors by definition lower the local reward rate) and on surprise. In our task, errors signal deviations from an expectation, as most trials are correct, and prediction errors are believed to be important for the recruitment of performance monitoring (*Notebaert et al., 2009*). In fact, similar brain systems appear to be recruited after mistakes and after surprising outcomes (*Wessel et al., 2012*).

A recent study characterized the role of a projection from the anterior cingulate cortex (ACC) to the visual cortex (VC) on performance monitoring in the mouse, showing that post-error increases in performance in a visual attention task can be mediated by this projection (*Norman et al., 2021*). Although the authors did not interpret their findings in the context of modulations in cortical state, there are interesting parallels between their results and our findings. Optical pulsatile activation (30 Hz) of the ACC to VC projection resulted in decreases in low-frequency LFP power (consistent with a decrease in our Synch measure, *Figure 1H1*) and increases in high-frequency power (consistent with an increase in FR; *Yizhar et al., 2011*; *Guyon et al., 2021*) in the VC – akin to our favorable state for accuracy after errors. Interestingly, a behavioral effect of either excitation or suppression of this projection was only observed when the manipulation was performed in the baseline period after errors, so the effect of the manipulation in *Norman et al., 2021* is also gated by previous-trial outcome. These findings suggest that the favorable state for accuracy after errors we identified might signal the successful recruitment of performance monitoring frontal networks (which for the auditory modality comprise the ACC and also premotor cortex [*Zhang et al., 2016*; *Sun et al., 2022*]). After correct trials, the link between baseline fluctuations and medial frontal areas might be weaker, or might be intact, but the corresponding top-down projections appear ineffective (*Norman et al., 2021*), which would explain the absent relationship between baseline fluctuations and choice accuracy that we see after correct trials. This outcome dependence of top-down influence, which potentially explains our results and those of Norman et al., suggests that errors produce changes in functional connectivity. The thalamus has recently been suggested to be important for this function (*Nakajima and Halassa, 2017*), and is also an important and necessary structure in the performance monitoring network (*Seifert et al., 2011*; *Peterburs et al., 2011*; *Ullsperger et al., 2014*), which projects to the ACC (*Seifert et al., 2011*). Changes in the activity of the local ACC network have also been suggested to gate functional connectivity between sensory and motor ACC ensembles (*Kim et al., 2021*). Finally, neuromodulatory systems, which are engaged by prediction errors and negative outcomes (*Hollerman and Schultz, 1998*; *Gardner et al., 2018*; *Fischer and Jocham, 2020*; *Danielmeier et al., 2015*) are likely to coordinate large-scale changes in brain-wide functional connectivity.

Although the effect of cortical fluctuations on both choice accuracy (*Figure 3*) and stimulus discriminability (*Figure 3—figure supplement 6*) changes after errors, the nature and range of cortical fluctuations themselves is only weakly affected by outcome (*Figure 4*). This suggests that cortical synchronization and FR are correlates of a number of distinct physiological processes. In fact, there is substantial evidence that this is the case, as cortical fluctuations are shaped by neuromodulation (mainly cholinergic [*Goard and Dan, 2009*; *Chen et al., 2015*; *Reimer et al., 2016*]), locomotion and arousal (*McGinley et al., 2015a*; *Vinck et al., 2015*) and, specially for the auditory cortex, motor activity (*Schneider et al., 2014*). The effect of spontaneous fluctuations on evoked sensory responses and on behavior is thus likely to be context dependent, reflecting the differential engagement of these diverse brain systems in different situations.

Our work complements previous characterizations of the role of brain state using detection GO-NOGO tasks. *McGinley et al., 2015a* found a non-monotonic effect of pupil size and synchronization on performance in an auditory detection paradigm for mice, a pattern often observed (*McGinley et al., 2015b*; *Yerkes and Dodson, 1908*; but see *Neske et al., 2019*). In studies using sensory detection tasks for human subjects, another consistent finding is a relationship between electroencephalogram (EEG) power in the alpha range and responsiveness (*Ergenoglu et al., 2004*; *Iemi et al., 2017*; *Samaha et al., 2020*) (subjects are less responsive when alpha power in the pre-stimulus baseline is higher). 2AFC discrimination tasks and GO-NOGO detection tasks, however, place different requirements on the subject and, in particular, differ on the extent to which variations in overall responsivity affect task performance. As such, it might be expected that the relationship between brain state and performance in these two types of psychophysical paradigms differs. Consistent with this idea, alpha

power in the pre-stimulus baseline consistently lacks association with choice accuracy in discrimination tasks (*Ergenoglu et al., 2004*; *Iemi et al., 2017*; *Samaha et al., 2020*). Nevertheless, *Waschke et al., 2019* found an inverted-U relationship between pupil size and choice accuracy in a 2AFC pitch discrimination task performed by human subjects, consistent with results in sensory detection tasks. *Beaman et al., 2017* found that monkeys were more accurate in a 2AFC delayed visual discrimination task if activity in the delayed period was more desynchronized and that, in this state, the comparison stimulus was more discriminable, consistent with our results after error trials (*Figure 3C*, *Figure 3—figure supplement 6*). The effect of cortical state on accuracy in this study was evident, however, regardless of the outcome of the previous trial. This could reflect the fact that primates, by default, engage cognitive control to solve the delayed visual discrimination task. *Waschke et al., 2019* did not find a positive relationship between auditory cortex desynchronization and discriminability (instead, they found an inverted-U relationship between desynchronization and bias). Regarding work in rodents, another recent study by *Jacobs et al., 2020* examined the relationship between cortical state fluctuations and performance in a 2AFC visual discrimination task for mice. Our results are consistent with theirs regarding the lack of effect of cortical state on accuracy when trial outcome is not considered (*Figure 3A*), but the outcome-dependent relationship between cortical state fluctuations and accuracy which we revealed was not addressed in this study.

Overall, our results and those from previous studies suggest that the relationship between brain state during the pre-stimulus baseline and performance is more subtle during discrimination than during detection. These studies, however, are heterogeneous, involving different species (few reports exist, for instance, on brain-state modulation of discrimination accuracy in 2AFC tasks in rodents) and methods for assessing brain state. Although we believe that measures of desynchronization based on absence of low-frequency (delta) power should be consistent with each other, whether derived from spikes, calcium imaging, LFP or EEG (*Whittingstall and Logothetis, 2009*; *Figure 1I*), different species and tasks, and even subtle task differences in the case of rodents, might lead to different results. *Jacobs et al., 2020*, for instance, found cortical desynchronization to be associated with engagement, whereas we did not (*Figure 5F*). This discrepancy might be due the different behavioral state of the mice in both studies before Skip trials. For our mice, facial movement (OpticF) during the baseline is a significant predictor of Skips (*Figure 5F*, *Figure 3—figure supplement 5H*), signaling that mice move their faces more than average while they are distracted and disengaged, whereas mice in the *Jacobs et al., 2020* study had a no-movement trial-initiation condition.

Although we found no relationship between baseline pupil size or synchronization and Skip probability (*Figure 5F*), both of these baseline signals where associated with the probability of a premature response, which were more likely when the baseline was more synchronized and the pupil was smaller (*Figure 5D*). This is interesting given that, in tasks without a delay period, it is pupil dilation (*McGinley et al., 2015a*) and desynchronization (*Jacobs et al., 2020*) that tend to be associated with faster RTs and 'false alarms'. On the other hand, the result is expected given the well-known association between pupil dilation and inhibitory control (*van der Wel and van Steenbergen, 2018*). In an anti-saccade task, for instance, it was found that pupil size was bigger before correct anti-saccades than before incorrect pro-saccades in anti-saccade trials (*Wang et al., 2015*). That a diversity of cognitive processes converge on pupil dilation is consistent with its dependence on different neuromodulatory systems (*Joshi et al., 2016*; *Reimer et al., 2016*; *Cazettes et al., 2021*). In tasks with a delay period, explanatory accounts of pupil dilation based on distractability or exploration (*Gilzenrat et al., 2010*; *Aston-Jones and Cohen, 2005*) and cognitive control (*Kahneman and Beatty, 1966*; *van der Wel and van Steenbergen, 2018*) appear to make opposite predictions regarding responsivity. In our task, processes associated with control seem to have a stronger hold on the pupil signal.

Our results, together with those from previous studies (*Jacobs et al., 2020*), demonstrate that mice can sustain high-level discrimination performance relatively independently of cortical synchronization (in our case after correct trials). What general conclusions can be derived from these findings regarding the relationship between cortical state and sensory discrimination accuracy? In addressing this question, we first note that, in humans, good levels of performance can be obtained in well rehearsed tasks, with high degrees of automaticity and in the presence of frequent feedback – exactly the conditions present in psychophysical tasks like ours – in the absence of the kind of mental effort associated with focused attention (*Harris et al., 2017*; *Gold and Ciorciari, 2020*). These 'flow' states, in which subjects experience dissociation and lack of self-consciousness, are thought to arise when

skills and demand are matched (*Csikszentmihalyi, 1990*). Interestingly, brain structures implicated in performance monitoring and engaged by task errors, such as the ACC and medial prefrontal cortex (*Botvinick et al., 2001*; *Ullsperger et al., 2014*; *Norman et al., 2021*), are downregulated during flow (*Ulrich et al., 2016*; *McGuire and Botvinick, 2010*). We hypothesize that, when discriminating simple sensory stimuli, mice might operate in a behavioral state equivalent to 'flow' during streaks of correct trials, with different brain systems sustaining performance in this state compared to the behavioral state prevalent after errors. Thus, unlike the hypothesis advanced in the introduction, good performance in sensory discrimination might not necessitate a state of elevated top-down control where cognition is oriented toward the environment, and might instead also be possible in flow-like states characterized by effortless automaticity. Our results suggest that, in these states, cortical state fluctuations in sensory areas are not relevant for accurate choices.

## Materials and methods

All procedures were reviewed and approved by the Champalimaud Centre for the Unknown animal welfare committee and approved by the Portuguese Direção Geral de Veterinária (Ref. No. 6090421/000/000/2019). All experiments were performed using male C57BL/6J mice that were housed on a 12-hr inverted light/dark cycle.

### Head bar surgery

During induction of anesthesia, animals (6–8 weeks of age, 20–22 g body weight) were anesthetized with 2–3% (volume in $O_2$) isoflurane (Vetflurane, Virbac) and subsequently mounted in a stereotactic apparatus (RWD Life Science) on a heating pad (Beurer). Once animals were stably mounted, isoflurane levels were lowered to 1–1.5% and the eyes were covered with ointment (Bepanthen, Bayer Vital). The head was shaved and the scalp cleaned with betadine. A midline incision was performed to expose lambda and bregma, which were subsequently used to align the skull with the horizontal plane of the stereotactic frame by measuring their position with a thin glass capillary (Drummond Scientific). The skull anterior of bregma was exposed by cutting a small area of skin. The exposed area was cleaned with betadine and slightly roughened by scraping it with a surgical blade (Swann-Morton). Subsequently, the skull was dried with sterile cotton swabs and covered with a thin layer of super glue (UHU). To further increase long-term stability, four 0.9-mm stainless steel base screws (Antrin Miniature Specialties) were placed in the skull. The exposed skull and base screws were then covered with dental cement (Tap2000, Kerr). A custom designed head bar (22 × 4 × 1 mm, aluminum, GravoPlot) was lowered into the dental cement while still viscous until the head bar was in contact with the base screws. Subsequently, an extra drop of dental cement was applied to the center of the head bar in order to fully engulf its medial part. The remaining skin incision along the midline was then sutured. The animals were injected with buprenorphine (opioid analgesic, 0.05 mg/kg) into the intraperitoneal cavity and allowed to recover for 3–5 days.

### Training

We adapted previously described procedures for training head-fixed mice in psychophysical tasks (*Guo et al., 2014*). After recovery from head bar implantation the animals were water deprived for 12 hr prior to the first handling session. In handling sessions, mice were accustomed to the experimenter and being placed in an aluminum tube to restrain their movement. In the first days of handling, the tube was placed in the animal's home cage. Once the mouse voluntarily entered the tube, it was presented with water delivered manually from a syringe at the end of the tube. This procedure therefore roughly mimicked the water delivery system in the training apparatus. Mice were allowed to drink a max of 1.5 ml of water during each handling session (30 min). Once mice were accustomed to receiving water in the aluminum tube and being handled by the experimenter, they were placed in the behavioral setup and head fixed with the two water delivery spouts approximately 1 cm in front of their mouth. To adapt them to head fixation, free water was delivered upon licking at either of the water delivery spouts. Lick detection was based on junction potential measures between the aluminum restraining tube and the stainless steel lick spout (*Hayar et al., 2006*). After triggering and consuming 15 rewards (single reward size: 3 µl), training proceeded as follows. In the first stage of training, every 3.1 s a random high (distribution: 22–40 kHz, presented sound randomly selected each

trial, category threshold at 14 kHz) or low (distribution: 5–8.5 kHz) frequency sound was presented to both ears at 60 dB SPL for 600ms, indicating at which of the spouts water was available (mapping is counterbalanced across animals). 150 ms after sound onset, a green LED flash of 50 ms indicated the onset of the 1.5-s response period. If the first lick in the response period occurred at the correct water spout, a 3-µl water reward was delivered. In order to facilitate the animals' engagement, a free water drop was delivered 150 ms after sound onset in a random 10% subset of trials. Once mice were readily trying to trigger water rewards by licking at either lick spout after sound presentation (minimum of 18 out of the last 20 trials without free water), the sound duration was reduced to 150 ms, followed by a 1-s response period. The ITI was drawn randomly from a set of four possible values: 3, 4, 5, and 6 s. After mice were engaged in the new timing of the task, free water delivery ceased and incorrect responses were punished by an additional 6-s time delay in between sound presentations. As soon as the animals had learned to correctly respond by licking at the appropriate water delivery spout in at least 34 of 40 consecutive trials, the response delay was introduced, by gradually delaying the appearance of the go signal. Impatient licks triggered the abort of the trial and were signaled with white light flashes. The delay period was increased in 10 ms increments as long as the animal performed at an accuracy of at least 80% for maximal five increments per session.

Mice were taught to withhold their responding after the stimulus by progressively delaying the go signal across sessions, contingent on their ability to refrain from premature responding. After this process, which typically took 10–12 weeks of training, the difficulty of the presented frequencies was gradually increased by approximating the range of possible low and high frequencies. These increases were performed in 19 increments, depending on a low bias and high performance (bias ≤20%; performance ≥80%, only one change per training session), until a final frequency distribution of low (9.9–13 kHz) and high (15–20 kHz) frequencies was reached. After reaching the final frequency distributions, mice were presented with three fixed frequencies per condition (low: 9.9, 12, and 13 kHz; high: 15, 16.3, and 20 kHz), with the easy conditions presented only 15% of the times to obtain more error trials and hard trials presented in 8% of the trials. Due to the resulting low number of hard trials per behavioral session, their presentation was omitted during most acute recording sessions (24 of 36). Excessive bias or disengagement at any time during the training were corrected by delivering free water at the unpreferred spout right after stimulus presentation until the animal readily responded again. All such intervention trials and the trials subsequent to each of them were excluded from analysis.

## Electrophysiological recordings

Six to 12 hr prior to the first probe insertion in each hemisphere, mice were deeply anesthetized with 2–3% (volume in $O_2$) isoflurane, mounted in a stereotactic apparatus and kept on a thermal blanket. The eyes were covered with ointment. Isoflurane levels were subsequently lowered to 1–1.5%. The animal's head was placed in a stereotactic frame using the head bar. The skin covering the areas above the recording sites and the midline was removed and the exposed skull was cleaned from periostium with a surgical scalpel blade and cleaned with betadine and dried with sterile cotton swabs. Subsequently a small craniotomy was performed above the desired recording site (2.8 mm posterior, 2.2 mm medio-lateral to bregma under a 35° medio-lateral angle). The exposed dura mater was opened using a small needle (BD Microlance 0.3 × 13 mm) and subsequently the recording silicone probe (BuzA64sp, Neuronexus) was slowly lowered to the desired depth (2.6 mm from brain surface). Probes were inserted with the shanks in a medio-lateral orientation, so that the six shanks in the final position approximately span the cortical layers (*Figure 1—figure supplement 1A*). Neural activity was digitized with a 64-channel headstage (Intan) at 16 bit and stored for offline processing using an Open Ephys acquisition board (Open Ephys) at a 30 kHz sampling rate. Behavioral sessions and storage of recording neural signals started only 10–20 min after probe insertion to allow for tissue relaxation and stabilization of the recording. Recording sessions were limited to three recordings per hemisphere in each animal due to the tissue damage caused by probe insertion. In the final recording session in each hemisphere the probe was coated with DiI (VybrantTM DiI, Invitrogen) to confirm correct placement of the recording probe histologically.

## Dataset

We recorded neural activity in 36 behavioral sessions from 6 mice (3 recordings per hemisphere). Out of these, 23 sessions had at least 100 trials and a behavioral sensitivity ($d'$ from signal detection

theory) of at least 1, and were considered for further analysis. From these 23 sessions, 3 sessions from 1 mouse were discarded from the dataset because it was not possible to properly estimate the size of its pupil due to eyelid inflammation. We confirmed histologically that recordings were made in both primary, as well as ventral and dorsal auditory cortex, using the location of the recordings shanks (during the last insertion of each hemisphere as a reference for all insertions at these target coordinates) relative to salient anatomical landmarks. Relevant brain structures for this purpose were identified by comparing the fluorescent patterns obtained in a DAPI staining with reference areas demarcated in *Paxinos and Franklin, 2007*. Thus, the correct probe placement along the anterior–posterior axes, as well as placement of all shanks within the cortex, could be confirmed reliably (see *Figure 1—figure supplement 1A* for an example recording). In hemispheres where the most medial shank exceeded cortical depth (1 of 12), this shank was excluded in all recordings performed in that hemisphere that passed all criteria for inclusion (2 of 20). Thus, the dataset consists of 18 recordings of 6 shanks, and 2 recordings of 5 shanks. For 6 out of these 20 recordings, we have histological verification of probe/shank placement.

Unless otherwise specified in the text, we did not consider for analysis the first 10 trials in each behavioral session during which the mice are adjusting to the setup and the position of the licking ports is being fine tuned. We also did not consider trials where the current or the previous trial were free rewards (trials in which the experimenter delivered a free reward to re-engage the animal). For the analysis on accuracy we considered the first lick within the response window (0.7–1.7 s after sound onset) which was also used to determine if animals would be rewarded. For the analysis of engagement, 'skips' were defined as trials in which no licks were detected in the first 2 s since stimulus onset. Premature responses were defined as trials in which the first lick occurred before the go-signal, at 0.65 s.

## Videos recording and analysis

We collected videos of mice performing the task at 60 fps using regular USB cameras without an infrared (IR) filter and applying direct IR illumination to increase pupil contrast (*Figure 1—figure supplement 1E*). From the videos, we extracted a proxy for face movement and one for arousal. For face movement, for each recording session, we selected a region of interest (ROI) around the face of the animal and computed the average magnitude of the optic flow in that ROI (using Lucas–Kanade method [*Lucas and Kanade, 1981*]). To compare across sessions, we *z*-scored the optic flow session by session. What is referred to as OpticF in the text corresponds to the median OF in the baseline period (2 s before stimulus presentation). As a proxy for arousal, we estimated pupil size. We used DeepLabCut (DLC; *Mathis et al., 2018*) to detect points around the pupil frame by frame and then estimated the pupil size as the major axis of an ellipse fitted using those points (for robustness of the pupil estimates, we further smoothed the data by applying a robust local regression using weighted linear least squares and a first degree polynomial model with a 250 ms window – rlowess in MATLAB). For training the model using DLC, we labeled 8 points in 20 frames for each of the 20 behavioral sessions. To remove frames where the detection was poor, we only considered those where the average likelihood of the DLC detection was higher than a threshold (0.8). Finally, for each session, we normalized the pupil by the 2% lowest values in the session (so, e.g., 100% means a 100% increase in pupil size relative to its smallest values). What we referred in the main text as PupilS represents the median values of the pupil in the baseline period (2 s before stimulus presentation).

## Spike sorting

Spike events were detected using Kilosort2 (*Pachitariu et al., 2016*; *Pachitariu, 2020*) and subsequently manual clustering was performed using phy2 (*Rossant, 2020*) to remove artifact clusters. We did not use unit identity in any of our analyses, which pertained only to the structure of the population ('multiunit' MUA) activity.

## Estimation of baseline FR and synchrony

We described baseline neural activity in each trial using two variables, the population FR and synchrony (Synch; *Figure 1F*). We estimated FR as the average number of spikes of the MUA in the baseline period (average in time and across the number of units). To estimate synchrony, we first pooled all spikes from the units in the baseline period. Then computed the magnitude of the standard deviation

across time of the instantaneous FR (in bins of 20 ms) of the population (which is a measure of the population averaged covariance between all pairs [*Renart et al., 2010*]) and divided it by the average of the same quantity calculated for 100 surrogates where the spike times of the MUA in that particular baseline are randomly shuffled (*Figure 1E*). We used this measure because we observed that it is less dependent on overall number of spikes in the baseline period than related measures such as the coefficient of variation of the MUA across time (*Renart et al., 2010*; *Kobak et al., 2019*). This measure of synchrony is 'normalized', with a reference value of 1 expected if the neural population is asynchronous and neurons fire with Poisson-like statistics. In *Figure 1G*, we assessed synchrony using the coefficient of variation of the MUA, defined as the ratio between the standard deviation and the mean the spike count of the MUA across each 20-ms period in the baseline period.

## Spectral analysis

We performed spectral analysis using the Chronux MATLAB Package (http://chronux.org). In particular, we used the function mtspectrumpt.m, which uses a multitaper approach to calculate efficiently the power spectrum of a point process. In *Figure 1H*, for each of the four example baseline periods, we used a value for the time-bandwidth parameter $TW = 10$. For *Figure 1I*, since additional smoothing is provided by the average across trials, we used $TW = 5$. In each case, we used the recommended $2TW - 1$ tapers to calculate the spectrum in each baseline period. Each power spectrum was normalized by the mean power for all frequencies above a high-frequency cutoff of 10 kHz (the sampling rate of the recordings was 30 kHz), which is equivalent to a normalization by the FR within that baseline period (since the high-frequency limit of the spectrum of a point process is the FR).

## Innovations

We 'cross-whitened' the four signals under analysis (FR, Synch, OpticF, and PupilS) by making linear fits of each of them separately for each session, using as regressors the outcome in the previous 10 trials (1: reward; 0: no reward), the values of four signals in the previous 10 trials and the *current trial number* (TrN, to account for within-session trends). Each regression thus specified 51 parameters plus the offset. We then defined the innovations FR$_I$, Synch$_I$, PupilS$_I$, and OpticF$_I$ as the residuals of this linear fit (*Figure 2—figure supplement 2*). In *Figure 4*, we address the influence of the outcome of the previous trial on the four baseline innovations. We did this by trying to explain previous outcome using a GLMM based on these signals plus the session trend. For this fit, the innovations were modified by excluding previous-trial outcome as a regressor (since their relationship to outcome is the target of the analysis).

## Generalized linear mixed models

To analyze the behavioral and neural data we used GLMM (*Stroup, 2013*) (using the function fitglme in MATLAB) using recording session as a random effect for both slopes and offset. When fitting continuous variables (e.g., FR$_I$ in *Figure 2D*) we used a linear mixed model. When fitting binary variables (such as accuracy or skips) we used a binomial distribution and a logit link function. In order to prevent global covariations between session-by-session differences in the marginal statistics of the predictors and the prediction targets to contribute to the trial-by-trial associations that we seek to reveal, we always *z*-scored all predictors separately *within each session*. In all fits, we also include a regressor with the number of the trial in the session (TrN) to account for session trends in the target of the fit. In *Figure 3C, E*, we evaluated the *joint* effect of FR$_I$ and Synch$_I$ on choice (rightmost predictor). To do this, we first constructed a joint predictor by projecting each *z*-scored (FR$_I$($z$), Synch$_I$($z$)) pair (for each trial) onto an axis with −45 deg slope for *Figure 3C* (so that the joint predictor would take large positive values when the baseline state was favorable after errors), or 45 deg slope for *Figure 3E* (using the same reasoning after corrects). We then run GLMMs in which the two separate FR$_I$ and Synch$_I$ predictors were replaced by the single joint one. In *Figure 3C, E*, we only show the value of the joint coefficient in these new GLMM fits. The values of all other predictors were not different. The specific models that we fitted to the data are the following (in Wilkinson notation). To predict FR and Synch in *Figure 2A*, we used the model $FR \sim 1 + OpticF + PupilS + (1 + OptiF + PupilS|session)$ and $Synch \sim 1 + OpticF + PupilS + (1 + OptiF + PupilS|session)$. For *Figure 2D*, we used the same model but using innovations, $FR_I \sim 1 + OpticF_I + PupilS_I + (1 + OpticF_I + PupilS_I|session)$ and

$Synch_I \sim 1 + OpticF_I + PupilS_I + (1 + OpticF_I + PupilS_I|session)$. The model we fitted to the accuracy of the mice (**Figure 3C, E**) is:

$$Correct \sim 1 + Stim + TrN + OpticF_I + PupilS_I + FR_I + Synch_I +$$
$$+ (1 + Stim + TrN + OpticF_I + PupilS_I + FR_I + Synch_I|session)\;.$$

To fit the model to the accuracy independently on the outcome of the previous trial (**Figure 3A**), we also included *pCorr* both as a fixed and random term. The model we fitted to the accuracy of the animals in the previous trial (**Figure 4E**) is:

$$pCorr \sim 1 + TrN + OpticF_I + PupilS_I + FR_I + Synch_I +$$
$$+ (1 + TrN + OpticF_I + PupilS_I + FR_I + Synch_I|session)\;.$$

The model we fitted to premature responses (**Figure 5D**) is:

$$Premature \sim 1 + TrN + pPrem + pCorr + pSkip + OpticF_I + PupilS_I + FR_I + Synch_I +$$
$$+ (1 + TrN + pPrem + pCorr + pSkip + OpticF_I + PupilS_I + FR_I + Synch_I|session)\;,$$

while the one for skips (**Figure 5F**) is:

$$Skip \sim 1 + TrN + pPrem + pCorr + pSkip + OpticF_I + PupilS_I + FR_I + Synch_I +$$
$$+ (1 + TrN + pPrem + pCorr + pSkip + OpticF_I + PupilS_I + FR_I + Synch_I|session).$$

In **Figure 3—figure supplement 5**, we applied the same models but without innovations for panels E–H while we also removed *TrN* for panels A–D. In **Figure 5—figure supplement 1**, we predict the RT of the animals with or without innovations using the same predictors we used for skips and premature responses.

Although our GLMMs often contained many predictors and their relative random slopes, they generally converged and gave consistent results across our resampling procedure (which we used for estimating CIs on the magnitude of the model coefficients). However, since we run many resamples (5000), sometimes the results were inconsistent. We identified this 'outlier' model runs as those for which the (absolute) distance between any of the GLMM coefficients in the model, and the median of the distribution across resamples, was more than seven times the MAD. These cases constituted just a small portion of all resamples. For instance, for predicting accuracy using all valid trials, the proportion of outliers was 0.12%, after error trials it was 0.22%, and after correct trials 0.08%. These outliers model runs were excluded from the statistics we used to report the results.

## Analysis of the sound-evoked neuronal activity

In order to determine if information in the evoked population activity about the stimulus or upcoming choice depends on properties of the pre-stimulus baseline, we devised a two-step analysis workflow.

In the first step, for every recording session, we used the evoked activity in each trial – defined as the number of spikes fired by each unit during the stimulus presentation (0–150 ms) – to decode stimulus or choice (the number of predictors is therefore equal to the number of units). Because performance, on average, is relatively high (approximately 80% correct), stimulus and choice are correlated. Thus, to make sure that choice decoding did not reflect tuning to the stimulus (and vice versa), we constructed separate choice decoders for each of the two stimulus categories (and vice versa). Specifically, for each recording we performed cross-validated (five folds) L2-regularized logistic regression on 90% of the data to identify decoders that could be used to predict choice/stimulus on the remaining 10% of the data. The same procedure was repeated for each fold (10 times). To reduce variability due to randomness in fold selection during cross-validation, the same procedure was repeated 1000 times and the projection that we used in the following part of the analysis was the median across the 1000 repetitions of this procedure. The two projections for choice (stimulus) conditional on stimulus (choice) were merged into a single 'projection' regressor that contained the projection onto the relevant axis for each trial. Logistic regression was performed using the MATLAB version of the free software package glmnet (**Friedman et al., 2010**) (http:// hastie.su.domains/glmnet_matlab/). In the second step, we used a GLMM to predict either stimulus category or choice with recording session as a random effect. The predictors for the analysis

were the scalar projection that we identified in the first step of the analysis, $FR_l$ and $Synch_l$ as in the rest of our study, and the interaction terms between the projection and $FR_l$ and $Synch_l$. Projections and $FR_l$ and $Synch_l$ were *z*-scored separately for each recording. The GLMM analysis could then be performed independently for after correct and after error trials. Weights of the model and 95% CI in *Figure 3—figure supplement 6* were found using the parametric estimation of the fitglme function in MATLAB.

## Statistics

We estimated the uncertainty of the estimates of the coefficients of our GLMM fits using bootstrap resampling (*Efron and Tibshirani, 1994*). We resampled with replacement 'hierarchically', so that the number of trials from each recording was preserved in each global surrogate. Distributions of the magnitude of each coefficient and associated 95% CIs came from 5000 resamples. In figures, we always display median, interquartile range and 95% CI for each coefficient. p values for the null hypothesis of a coefficient being equal to zero were computed using the quantile method (*Efron and Tibshirani, 1994*), that is twice the value of the fraction of resamples with opposite sign as the estimate of the coefficient from the data. For consistency, we verified that estimates of significance obtained using bootstrap CIs for parameters agreed with parametric estimates from fitglme (*Figure 3—figure supplement 1*) which uses an approximation to the CMSEP method (*Booth and Hobert, 1998*). To test for differences in accuracy after a correct versus an error trial (*Figure 4B*), we computed, for each recording, the difference between the median accuracy of trials where the previous trial was correct and the median accuracy of trials where the previous trial was an error. We assessed the significance of this difference using a Wilcoxon signed rank test. To fit psychometrics curves, we used the psignifit MATLAB toolbox (*Schütt et al., 2016b*; *Schütt, 2016a*). When fitting an aggregate psychometric across sessions, we weighted each trial by the proportion of trials its corresponding session contributes to the whole dataset. To test for differences in the slope of the psychometric functions in *Figure 4D, F* conditional on whether the baseline state was favorable or unfavorable, we used the difference in slope between fits of the aggregate data conditional on the state of the baseline as a test statistic. To assess the significance of this difference, we first computed the distribution of the test statistic under a null hypothesis of no difference implemented by randomly shuffling, within each session separately, the label that signals whether the baseline for a trial is favorable or unfavorable, and we then computed the fraction of the surrogates from this distribution for which the value of the test statistic was equal or larger than in the actual observed data. Unless otherwise stated, data across recordings are reported as median ± MAD.

## Acknowledgements

We thank Julien Fiorilli for help developing the lick detection hardware, the Vivarium and Hardware scientific platforms at Champalimaud Research for support, Fanny Cazettes and Zach Mainen for developing the task in *Figure 1—figure supplement 1J*, and Leopoldo Petreanu, Michael Orger, Jaime de la Rocha, and Tiffany Oña for comments on the manuscript. DR was supported by a Fundação para a Ciência e Tecnologia postdoctoral fellowship (SFRH/BPD/119737/2016) and a Marie Skłodowska-Curie postdoctoral fellowship (H2020-MSCA-IF-2016 753819), RS was supported by a doctoral fellowships from the Fundação para a Ciência e a Tecnologia. AR was supported by the Champalimaud Foundation, a Marie Curie Career Integration Grant PCIG11-GA-2012–322339, the HFSP Young Investigator Award RGY0089, and the EU FP7 grant ICT-2011-9-600925 (NeuroSeeker).

## Additional information

### Funding

| Funder | Grant reference number | Author |
|---|---|---|
| Fundação para a Ciência e a Tecnologia | Postdoctoral fellowship SFRH/BPD/119737/2016 | Davide Reato |

| Funder | Grant reference number | Author |
|---|---|---|
| H2020 Marie Skłodowska-Curie Actions | Postdoctoral fellowship H2020-MSCA-IF-2016 75381 | Davide Reato |
| Fundação para a Ciência e a Tecnologia | Doctoral fellowships | Raphael Steinfeld |
| Champalimaud Foundation | | Alfonso Renart |
| Marie Curie Career Integration Grant | PCIG11-GA-2012-322339 | Alfonso Renart |
| Human Frontier Science Program | Young Investigator Award RGY0089 | Alfonso Renart |
| Seventh Framework Programme | ICT-2011-9-600925 | Alfonso Renart |

The funders had no role in study design, data collection, and interpretation, or the decision to submit the work for publication.

## Author contributions

Davide Reato, Conceptualization, Resources, Data curation, Software, Formal analysis, Funding acquisition, Validation, Investigation, Visualization, Methodology, Project administration, Writing – review and editing; Raphael Steinfeld, Conceptualization, Resources, Data curation, Software, Validation, Investigation, Visualization, Methodology, Project administration, Writing – review and editing; André Tacão-Monteiro, Resources, Investigation, Methodology; Alfonso Renart, Conceptualization, Resources, Formal analysis, Supervision, Funding acquisition, Validation, Investigation, Visualization, Methodology, Writing - original draft, Project administration, Writing – review and editing

## Author ORCIDs

Davide Reato ⬤ http://orcid.org/0000-0001-5362-4616
Alfonso Renart ⬤ http://orcid.org/0000-0001-7916-9930

## Ethics

All procedures were reviewed and approved by the Champalimaud Centre for the Unknown animal welfare committee and approved by the Portuguese Direção Geral de Veterinária (Ref. No. 6090421/000/000/2019).

## Decision letter and Author response

Decision letter https://doi.org/10.7554/eLife.81774.sa1
Author response https://doi.org/10.7554/eLife.81774.sa2

# Additional files

## Supplementary files

• Supplementary file 1. We report in a table the statistics associated to the fixed coefficients in each of the generalized linear mixed models described in the main text. Starting from the left, each column represents: the figure in the text where the results are displayed, the prediction target, the predictors (one row per predictor), the median and lower and upper limits of the 95% confidence interval (Methods), the associated bootstrap p value (Methods), and the total number of observations (number of rows in the predictor matrix) in the model.

• MDAR checklist

## Data availability

All data and code necessary to reproduce the main findings of this manuscript are deposited on Dryad (https://doi.org/10.5061/dryad.w0vt4b8vf).

The following dataset was generated:

| Author(s) | Year | Dataset title | Dataset URL | Database and Identifier |
|---|---|---|---|---|
| Reato D | 2022 | Response outcome gates the effect of spontaneous cortical state fluctuations on perceptual decisions | https://dx.doi.org/10.5061/dryad.w0vt4b8vf | Dryad Digital Repository, 10.5061/dryad.w0vt4b8vf |

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
