## [Editor Report]

Reato and colleagues investigated a question that has long puzzled neuroscientists: what features of ongoing brain activity predict trial-to-trial variability in responding to the same sensory stimuli? The data demonstrate that taking into account behavior on the previous trial (specifically an incorrect choice) allows these associations to be seen. This is an important advance in our understanding of the relationship between brain state, behavioral state, and performance. Technically, the study is convincing, with appropriate and validated methodology in line with current state-of-the-art.

---

## [Decision Letter]

**Decision letter after peer review:**

Thank you for submitting your article "Response outcome gates the effect of spontaneous cortical state fluctuations on perceptual decisions" for consideration by *eLife*. Your article has been reviewed by 2 peer reviewers, and the evaluation has been overseen by a Reviewing Editor and Joshua Gold as the Senior Editor. The reviewers have opted to remain anonymous.

Essential revisions:

1) The concerns over data quality in repeated same-place recordings and the concerns over spatial specificity (see esp Rev 1) need to be addressed.

2) Concerns over the limited 'dynamic range' of arousal states samples (see esp. Rev 2) need to be addressed: To quote from Rev 2, 'To examine the full range of arousal states, it needs to be demonstrated that animals are varying between near-sleep (e.g. drowsiness) and high-alertness such as in rapid running. The interpretation of the results, therefore, must be taken in light of the trial structure and the states exhibited by the mice.' We will have to leave it to the authors at this stage, in how far additional data and/or analyses will be available to alleviate these interpretational concerns, or whether – at the very least – a more nuanced/restricted interpretation of the findings will be put forward.

*Reviewer #1 (Recommendations for the authors):*

– Inserting the probe 3 times into exactly the same coordinate (presumably on consecutive days – please add this) may cause much poorer recording quality on later days, as some neural death and gliosis will take place (indeed, supplementary figure 1a shows some loss of superficial tissue). could you show some evidence that each session had a sufficient quality of neural data and resulting spike sorting? and is the same true for behavior (how did performance and RT change over the 3 recording sessions)?

– To better compare these findings to ones from LFP/EEG, what would Figure 1H look like when computing power spectra of the LFP signal?

– l. 252-254, are these identical (please test this statistically)? I do not understand the author's rationale for concluding that they are 'almost identical'.

– What is the spatial specificity of these findings? The authors report recording from broad areas of auditory cortex, but then group all neurons together (and show histology for only one example animal). Is there a specificity to these effects wrt primary/secondary auditory areas or cortical layers, or would these effects be expected to be globally distributed? See e.g. https://doi.org/10.1101/2022.05.09.491042 for examples of how variation in placement of recording probes can affect electrophysiological measurements.

*Reviewer #2 (Recommendations for the authors):*

This is a nice study that examines the relationship between brain state, behavioral state, and performance. The efforts to look at real local interactions are well appreciated by this reviewer. However, the authors should be circumspect in the fact that their results are unique to their study and the parameters that they chose (and the states that the mice exhibited).

For example, the trials appear to be occurring rather rapidly. This means that the animal barely has the chance to stop licking to the previous trial before a new trial has begun. Thus, there will be a lot of interactions between the previous trial and the upcoming trial. The authors should try a trial structure that is much farther apart, allowing the trials to occur more independently. This will also change the state of the animal. Animals that are bombarded with trials every few seconds tend not to relax and therefore the study is likely to lack in mice with pupil diameters indicative of drowsiness or near-sleep (something I sometimes see in students in class!). It is hard to compare the current results with other studies, which may have worked hard to have animals that exhibit a wide range of states and which may have had a different trial structure.

The authors should at least discuss this point in their discussion and admit that their results are limited to their situation and to their mice.

---

## [Author Response]

Reviewer #1 (Recommendations for the authors):– Inserting the probe 3 times into exactly the same coordinate (presumably on consecutive days – please add this) may cause much poorer recording quality on later days, as some neural death and gliosis will take place (indeed, supplementary figure 1a shows some loss of superficial tissue). could you show some evidence that each session had a sufficient quality of neural data and resulting spike sorting? and is the same true for behavior (how did performance and RT change over the 3 recording sessions)?

We have clarified the recording procedure in the methods section (lines 696-707). In order to explore possible trends in behavior and neural activity as a function of session within the recording streak, we quantified, for each session, the number of units, the accuracy of the animals, and the median reaction time. We found that there was no specific trend with recording session in either of these three measures (Kruskal-Wallis one-way analysis-of-variance-by-ranks test, *p*_unitsD1−3_ = 0.23, *p*_unitsD4−6_ = 0.51, *p*_accuracy_ = 0.09, *p*_RT_ = 0.32). This information are now added in the text (lines 38-42) and in Figure 1—figure supplement 1.

– To better compare these findings to ones from LFP/EEG, what would Figure 1H look like when computing power spectra of the LFP signal?

Visual inspection of the raw voltage traces revealed the presence of artifacts in the low-frequency component of the extracellular signal caused by movement. To circumvent this difficulty, we reasoned that we could use our estimate of the facial movement of the animal (the optic flow of the video) to consider only trials where there was minimal movement in the pre-stimulus period that we used to determine the state of the cortex. We used a multi-taper approach (T W parameters [5 9]) to estimate the power on the LFP in the baseline period (the last two seconds before the presentation of the stimulus) and considered its average in the 4-16 Hz range. To focus in no-movement epochs we selected trials in the 2.5 percentile for OpticF. We found that low-frequency LFP power significantly correlated with our Synch estimate in these trials (t-test for significance of regression coefficients, *p* = 0.00009 adjusted for multiple comparisons using False discovery rate (FDR); Author response image 1 panel A). In contrast, the same analysis performed considering the 50 percentile did not show a significant relationship (t-test, *p* = 0.25, FDR; Author response image 1 panel B). Consistent with these results, the wider the range of movement allowed in the baseline (the larger the percentile), the weaker the correlation between low-frequency LFP power and our Synch measure (Author response image 1 panel C). To assess the relationship between Synch and the frequency content of the LFP, we computed (for the fraction of trials in the 2.5 percentile of OpticF) the ratio of power in the 4-16 Hz and 40-200 Hz ranges in each trial and examined whether it varied depending on the Synch level in the baseline. The ratio of low-to-high frequency power grows with Synch (mixed model; regression coefficient significantly greater than zero; *p* = 0.026; Author response image 1 panel D), suggesting that the Synch measure reflects more closely low-frequency power in the LFP. We conclude that our measure of local synchronization reflects global coordination of neural activity in a way similar to the way spectral analysis of the LFP does. This result is now mentioned in lines 80-85 of the text.

**Author response image 1. sa2fig1:** (A) Relationship between the LFP power (4-16 Hz) and the Synch measurement used to estimate the synchrony of the neuronal population in the pre-stimulus periods ([-2 0] s). Only the trials with the smallest facial movements were considered (2.5% of the trials with the lowest OpticF values). (B) Same as A but considering the 50% trials with the smaller movement. (C) R2 of the regression of LFP power in the 4-16 Hz range on our Synch measure as a function of the percentile of trials used from the baseline facial movement distribution. The more movement, the more the relationship between the LFP and Synch is obscured. (D) Power ratio between low (4-16 Hz) and high (40-200 Hz) frequency components of the LFP as a function of neural synchrony (for the fraction of trials in the lowest 2.5 percentile of baseline OpticF).

– l. 252-254, are these identical (please test this statistically)? I do not understand the author's rationale for concluding that they are 'almost identical'.

Thanks for pointing this out. Our language was not precise. We rewrote this sentence, stating that the predictive power of each regressor is qualitatively similar when using the raw predictors and their corresponding innovations.

– What is the spatial specificity of these findings? The authors report recording from broad areas of auditory cortex, but then group all neurons together (and show histology for only one example animal). Is there a specificity to these effects wrt primary/secondary auditory areas or cortical layers, or would these effects be expected to be globally distributed? See e.g. https://doi.org/10.1101/2022.05.09.491042 for examples of how variation in placement of recording probes can affect electrophysiological measurements.

The difficulty in defining the spatial specificity of our findings within the cortex is that, as we now clarify in the Methods section (lines 696-707), we only have histology from a subset of our recording sessions, since we record from the same mouse on several days (up to 6). Out of the 20 recording sessions we analyze, we only have histology from 6. And if we want to further split that subset of recording sessions according to probe location, we will be severly underpowered for the type of analysis we perform. However, as the reviewer points out, existing data suggests that patterns of cortical fluctuation tend to be global. For instance, Jacobs et al. (ref. 21) found, using wide field imaging, that behavior-related changes in cortical state (measured using the lowfrequency power of the GCaMP signal) on a visual task where evident across the dorsal cortex, even outside visual areas. Thus, we would hypothesize that baseline fluctuations in the primary versus surrounding auditory cortex would be related to discrimination accuracy in a similar manner to what we report on aggregate. However, due to the limitation just described, we cannot test this hypothesis directly. We now comment on the issue of the spatial specificity in the Discussion section (lines 433-439).

In order to assess the laminar specificity of our findings, we used the fact that our insertion strategy places all the shanks of the probe in a coronal plane, with each shank approximately parallel to the cortical layers (Figure 1—figure supplement 1). Although, again, there is variability in the exact dorso-ventral placement of the probe in each recording, the most dorsal shanks tend to record from neurons in more superficial layers and viceversa. Thus, we created putative superficial (deep) neural populations by selecting neurons recorded from the 3 most dorsal (ventral) shanks in each recording, and we estimated firing rate and synchrony in the pre-stimulus baseline separately for these two populations (fraction of superficial neurons relative to the total 0.55 ± 0.08, median ± MAD). Performing the same GLMM analysis reported in Figure 3 of the manuscript, we found that the general pattern of results in both putative subpopulations is similar to that found on aggregate, although the predictive power of FR_I_ and Synch_I_ when the analysis is done using each population separately are weaker. We hypothesize that this is due to the fact that the overall activity level and global synchrony calculated using populations of approximately half the size, provide less reliable estimates of the underlying cortical state. The results of this analysis are mentioned in lines 232-241 of the text, and in Figure 3—figure supplement 2C,D.

Reviewer #2 (Recommendations for the authors):This is a nice study that examines the relationship between brain state, behavioral state, and performance. The efforts to look at real local interactions are well appreciated by this reviewer. However, the authors should be circumspect in the fact that their results are unique to their study and the parameters that they chose (and the states that the mice exhibited).For example, the trials appear to be occurring rather rapidly. This means that the animal barely has the chance to stop licking to the previous trial before a new trial has begun. Thus, there will be a lot of interactions between the previous trial and the upcoming trial. The authors should try a trial structure that is much farther apart, allowing the trials to occur more independently. This will also change the state of the animal. Animals that are bombarded with trials every few seconds tend not to relax and therefore the study is likely to lack in mice with pupil diameters indicative of drowsiness or near-sleep (something I sometimes see in students in class!). It is hard to compare the current results with other studies, which may have worked hard to have animals that exhibit a wide range of states and which may have had a different trial structure.The authors should at least discuss this point in their discussion and admit that their results are limited to their situation and to their mice.

While we of course agree that care should be used in assessing the generality of the results from any particular study, the reviewer is suggesting that the time intervals between trials in our task are unusually short, that this will make it difficult for us to sample low arousal states, and that this should be highlighted in considering the generality of our findings.

Our ITIs vary between 3 and 6 seconds after correct trials and between 9 and 12 seconds after errors. Comparing these numbers with those used in studies linking brain state with psychophysical behavior, shows that our ITIs are not, in fact, unusually short. For instance, McGinley, McCormick and colleagues, in their classical 2015 publication (Ref. 10) used an ITI of 1 s in all trials except after false alarms, where the ITI was 8 s (average stimulus duration was approximately 4 s). Subsequent work in David McCormick’s lab has continued to use short ITIs. For instance, in Neske et al., 2019 (Ref. 19), ITIs were 2 s long, except after misses, where they were 4 s. Tasks studying brain state in McGinley’s lab also use shorter ITIs than ours (e.g., Gee and McGingley Strategic self-control of arousal boosts sustained attention, bioRxiv 2022, use a uniform ITI of 2-3 seconds, with a mean stimulus duration of 5 s). Jacobs et al. (Ref. 21) use the task from the international brain laboratory (IBL) task, which is self-paced. In this task, mean ITIs are 3.2 s and 4.0 s after correct and error trials respectively (I. Laranjeira, personal communication). In humans, Waschke et al., 2019 (Ref. 20), report a mean ITI of 9.14 s. This survey suggests to us that the ITIs in our task are not unusually short. Our minimum ITI is longer than that used in canonical papers, and our maximum ITI is longer than the one used in many of these studies, which suggests that we should be able to sample low pupil diameters and, more generally, that our task is not expected to produce a particularly narrow distribution of cortical states (at least not narrower than those studied in most other publications on this topic).

In order to quantitatively assess whether this is the case, we used two different strategies. First, we quantified the dependence of pupil size and our two measures of cortical state (FR and Synch) on ITI. Second, we compared the distribution of pupil sizes between our dataset and the dataset of our previous publication (Cazettes, Reato et al. (2021), ref. 40), which describes a self-paced task where mice run on a treadmill, and where mice voluntarily initiate trials, so that there are periods with long pauses between trials.

First, initial inspection of our pupil data during the baseline period reveals strong variations across trials (Figure 1—figure supplement 1F,G). We quantify pupil size in the *i^th^* frame within each session, as % increase relative to the minimum size (in pixels) in that session (specifically the median of the 2.5 % frames with smaller pupils across the session; Methods) i.e.,

Pupil\ sizei(%)= pupili−min[pupili]min[pupili] 100

where pupil*_i_* is the pupil size (long diameter of a fitted ellipse to the pupil in pixels) of the *i^th^* frame. For each baseline period, we compute the median Pupil size (%) across the frames in the two-second baseline period (see Methods). Pupil size displayed strong variations across trials in our dataset, with changes in diameter up to approximately 75% within sessions (Figure 1—figure supplement 1G).

ITI-dependence of pupil size and cortical state. We evaluated how the baseline pupil signal and measures of cortical state varied across ITIs. The distributions of baseline pupils are highly overlapping (Figure 1—figure supplement 1H, left). As expected, the mean of the distribution decays by ∼ 10 % (relative to min[pupil*_i_*]) as ITIs grow (Figure 1—figure supplement 1H, right), because shorter ITIs are associated with the previous trial being correct, in which case there is licking (and thus pupil dilation) during the ITI. For all ITIs, the distribution of the pupil had at least 25% of its probability mass for pupil sizes lower than 20% of its minimum within-session value (Figure 1—figure supplement 1H right), suggesting that constricted pupils are broadly sampled in our task regardless of the ITI. Furthermore, the range of the pupil size distribution was constant for ITIs longer than 9 s (quantile linear regression of 2.5 and 97.5 percentiles of the pupil against ITI; 95% CI_2.5_=[-0.32 0.54]; 95% CI_97.5_=[-0.03 1.34]; bootstrap), implying that the pupil size distribution reaches its steady state at ITIs of approximately 10 s and that longer ITIs will not lead to further changes.

Repeating the same analysis for FR and Synch, we see that the distribution of these measures of cortical state during the baseline period does not show any clear trend with ITI (Figure 1—figure supplement 1I), suggesting that the ITI does not play a large role in shaping the range of cortical states that we sample.

Comparison of pupil dilation across tasks: Panel J shows the distribution of pupil sizes in the Cazettes, Reato et al. dataset (Ref. 40) aggregated across mice (wildtype only), sessions and frames, processed in exactly the same way as outlined above, together with the corresponding distribution for our dataset (both for all frames, and for the baseline period). The main difference between the tasks is that probability mass in the intermediate quantiles of the distribution for the data in Cazettes, Reato et al. is shifted towards higher quantiles relative to our task (Figure 1—figure supplement 1J; the 95*^th^* percentile in our task is 58% and the one in the treadmill task is 77%). This is to be expected given that their mice spend a good proportion of the time running in a treadmill and that the task involves exploring the environment for extended periods of time by licking. We highlight, however, that the two distributions completely overlap in the lower end of the distribution of pupil sizes (marked with a yellow background in Figure 1—figure supplement 1J). This suggests that the pupil dynamics near the constricted end is very similar across the two tasks. Because the task in Cazettes, Reato et al. is self-paced, there are sometimes periods of immobility which correspond to pupil sizes in this range. This data provide further evidence that our task is not lacking in periods of constricted pupil. If anything, these results suggest that making our pupil size distributions more similar to those observed in tasks with locomotion would require including even shorter ITIs, not longer ones.

Because measures of pupil size are relative to the minimum across the session, we inspected example frames of the video to examine pupil sizes relative to the eye (insets, Figure 1—figure supplement 1J). This revealed that pupils in our dataset are, within each quantile, somewhat larger than in the Cazettes, Reato et al. dataset. This is likely due to the different lighting conditions across the two studies. While in both tasks infrared lights are used to illuminate the pupil, our mice worked in a closed behavioral box with only reduced illumination (precisely to avoid excessive pupil contraction due to ambient light), whereas the mice in the Cazettes, Reato et al. dataset worked in an open Faraday cage, in the presence of room lighting, and with a stripe of LEDs visible to the animal (to prevent animals from noticing optogenetics stimulation and avoid the stimulation to alter the ambient illumination and directly affect pupil size). Thus, the conditions of our experiment are rather well suited to explore the regime where contracted pupils reflect lack of arousal, rather than high ambient light.

Overall, this set of analyses suggest that, while the distribution of pupil sizes is quantitatively affected by both task structure (TS) and ITI duration, TS and ITI explain a small fraction of the variance in the pupil size distribution (and ITI explains essentially no variance of the FR and Synch distributions), which remains broad and with substantial trial-to-trial variability for all conditions that we have tested. ITIs in our task are neither particularly short compared to those used by other laboratories, and our longer ITIs are sufficiently long to allow the pupil to reach its steady state distribution. The lighting conditions in our experiment are adequate to study the connection between pupil and arousal specially at low arousal levels, and the distribution of pupil sizes in our task in the low-arousal regime is quantitatively very similar to that observed in a self-paced task that includes long periods between trials.

Thus, while the robustness of our results (regarding the relationship between cortical state and discrimination accuracy) against changes in task conditions is an empirical question which will have to be determined by future experiments, the analysis of pupil and cortical state across TS and ITI duration does not provide clear evidence in favor of the reviewer’s intuition that our ”results are unique to their study and the parameters they chose (and the states that the mice exhibited).”

We have added these results to Figure 1—figure supplement 1, and have added a short mention of this issue in the Results section (lines 86-88), and a longer paragraph in the Discussion section (lines 440-462) on the effect of task structure and ITI on measures of cortical state and arousal, in the context of the generality of our findings.